# Functional mapping of yeast genomes by saturated transposition

Agnès H Michel[1], Riko Hatakeyama[2], Philipp Kimmig[1], Meret Arter[1], Matthias Peter[1], Joao Matos[1], Claudio De Virgilio[2], Benoît Kornmann[1]*

[1]Institute of Biochemistry, ETH Zurich, Zurich, Switzerland; [2]Department of Biology, University of Fribourg, Fribourg, Switzerland

**Abstract** Yeast is a powerful model for systems genetics. We present a versatile, time- and labor-efficient method to functionally explore the *Saccharomyces cerevisiae* genome using saturated transposon mutagenesis coupled to high-throughput sequencing. SAturated Transposon Analysis in Yeast (SATAY) allows one-step mapping of all genetic loci in which transposons can insert without disrupting essential functions. SATAY is particularly suited to discover loci important for growth under various conditions. SATAY (1) reveals positive and negative genetic interactions in single and multiple mutant strains, (2) can identify drug targets, (3) detects not only essential genes, but also essential protein domains, (4) generates both null and other informative alleles. In a SATAY screen for rapamycin-resistant mutants, we identify Pib2 (PhosphoInositide-Binding 2) as a master regulator of TORC1. We describe two antagonistic TORC1-activating and -inhibiting activities located on opposite ends of Pib2. Thus, SATAY allows to easily explore the yeast genome at unprecedented resolution and throughput.

*For correspondence: benoit.kornmann@bc.biol.ethz.ch

**Competing interests:** The authors declare that no competing interests exist.

## Introduction

*Saccharomyces cerevisiae* is an invaluable model for cell biology (*Weissman, 2010*). Despite the simplicity of its genome, its inner working mechanisms are similar to that of higher eukaryotes. Furthermore, its ease of handling allows large-scale screenings. Yeast genetic screens have classically been performed by random mutagenesis, followed by a selection process that identifies interesting mutants. However elegant the 'tricks' implemented to expose the sought-after mutants, this selection phase remains a tedious process of finding a needle-in-a-haystack (*Weissman, 2010*). The selection phase can limit the throughput and the saturation of classical yeast genetic screens.

To circumvent these problems, a second-generation genetic screening procedure has been developed. Ordered deletion libraries for every non-essential gene have been generated (*Giaever et al., 2002*). The growth of each individual deletion strain can be assessed, either by robot-mediated arraying, or by competitive growth of pooled deletion strains, followed by detection of 'barcodes' that identify each deletion strain. These second-generation approaches also have limitations. First, ordered libraries of complete deletions only cover non-essential genes. Second, deletion strains are prone to accumulate suppressor mutations (*Teng et al., 2013*). To alleviate these problems, deletion libraries can be propagated in a diploid-heterozygous form. Additional steps are then required to make them haploid. In addition, while single genetic traits can be crossed into a pre-existing library, allowing for instance pairwise genetic-interaction analysis (*Costanzo et al., 2010*), introducing multiple and/or sophisticated genetic perturbations becomes problematic, since crossing requires a selection marker for each important trait. Typically, deletion libraries are missing in most biotechnology-relevant backgrounds. Finally, manipulating ordered libraries requires non-standard equipment, such as arraying robots, limiting the pervasiveness of these approaches.

**eLife digest** Genes are stretches of DNA that carry the instructions to build and maintain cells. Many studies in genetics involve inactivating one or more genes and observing the consequences. If the loss of a gene kills the cell, that gene is likely to be vital for life. If it does not, the gene may not be essential, or a similar gene may be able to take over its role.

Baker's yeast is a simple organism that shares many characteristics with human cells. Many yeast genes have a counterpart among human genes, and so studying baker's yeast can reveal clues about our own genetics. Michel et al. report an adaptation for baker's yeast of a technique called "Transposon sequencing", which had been used in other single-celled organisms to study the effects of interrupting genes. Briefly, a virus-like piece of DNA, called a transposon, inserts randomly into the genetic material and switches off individual genes. The DNA is then sequenced to reveal every gene that can be disrupted without killing the cell, and remaining genes are inferred to be essential for life.

The approach, named SATAY (which is short for "saturated transposon analysis in yeast"), uses this strategy to create millions of baker's yeast cells, each with a different gene switched off. Because the number of cells generated this way vastly exceeds the number of genes, every gene will be switched off by several independent transposons. Therefore the technique allows all yeast genes to be inactivated several times in one single experiment. The cells can be grown in varying conditions during the experiment, revealing the genes needed for survival in different situations. Non-essential genes can also be inactivated beforehand to uncover if any genes might be compensating for their absence.

In the future, this technique may be used to better understand human diseases, such as cancer, since many disease-causing genes in humans have counterparts in yeast.

Recently, an innovative approach called Transposon sequencing (Tn-seq) was developed in various bacterial models (*Christen et al., 2011*; *Girgis et al., 2007*; *van Opijnen et al., 2009*), and in the fungus *Schizosaccharomyces pombe* (*Guo et al., 2013*). By allowing *en masse* analysis of a pool of transposon mutants using next-generation sequencing, this strategy eliminates the drawbacks of previous genetic screens.

Here, we describe an adaptation of the Tn-seq strategy for *S. cerevisiae*, that combines the advantages of both first and second generation screens, while alleviating their limitations. The method is based on the generation of libraries of millions of different clones by random transposon insertion (*Figure 1A*). Transposons inserted in genes that are important for growth kill their hosts and are not subsequently detected. These genes therefore constitute transposon-free areas on the genomic map. Transposon-based libraries can be grown in any condition to reveal condition-specific genetic requirements. Unlike ordered deletion libraries, transposon-based libraries can easily be generated de novo from different strain backgrounds, are not limited to coding sequences and do not require the usage of robots.

This method can successfully uncover sets of genes essential in given conditions, genome-wide genetic interactions and drug-targets. Transposon insertions can generate loss- and gain-of-function variants. Finally, our approach not only shows which protein is important for growth, but also which part of the protein is essential for function, allowing genome-wide mapping of structural protein domains and screening of phenotypes at a sub-gene resolution.

## Results

### Library generation

The detailed procedure can be found in the Materials and methods section. The method utilizes the Maize Ac/Ds transposase/transposon system in yeast (*Lazarow et al., 2012*; *Weil and Kunze, 2000*). Briefly, cells in which the *ADE2* gene is interrupted by the MiniDs transposon are induced to express the transposase Ac, on galactose-containing adenine-lacking synthetic defined medium (SD +galactose -adenine). Transposase-induced MiniDs excision is followed by repair of the *ADE2* gene.

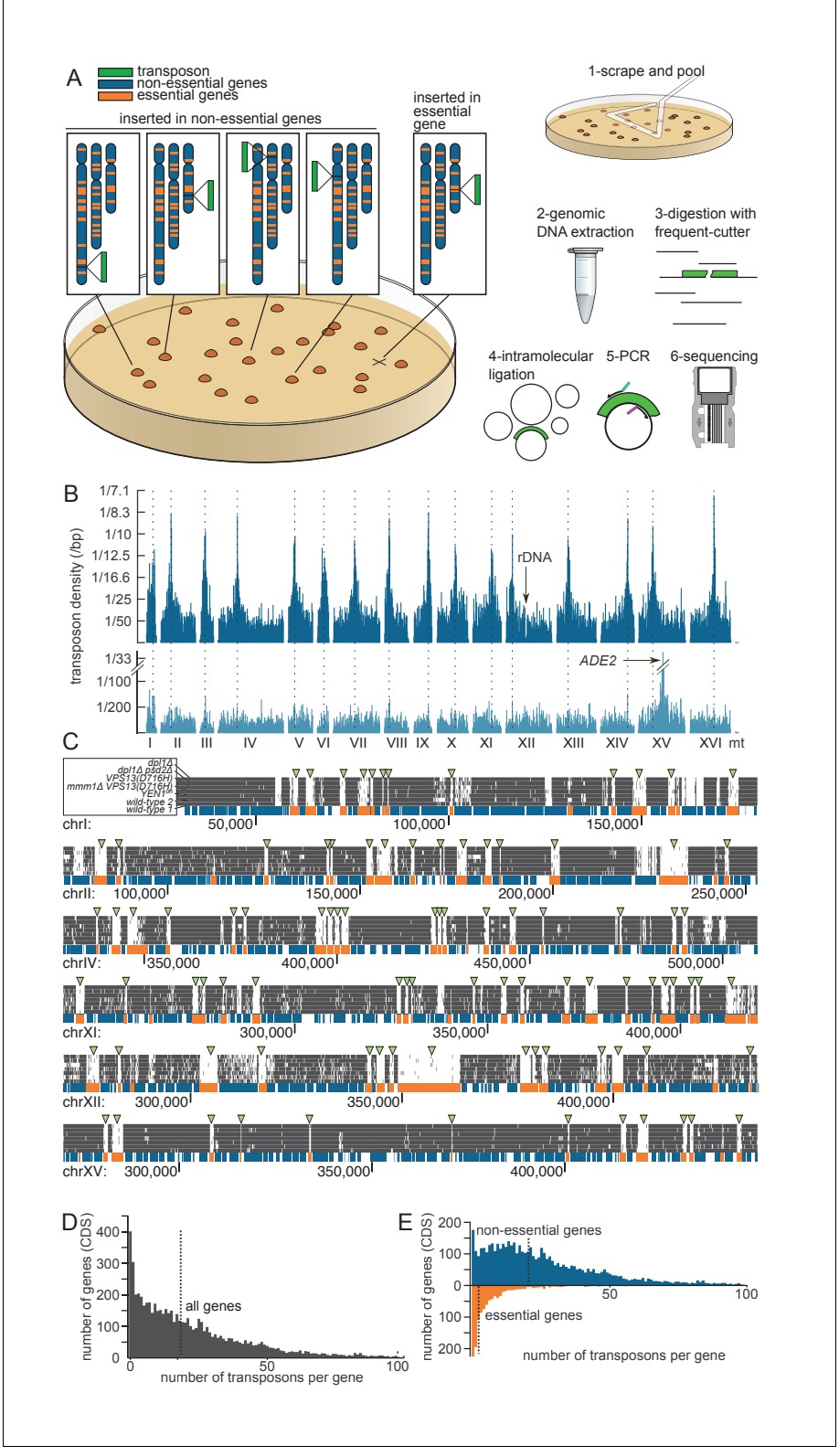

**Figure 1.** Principle of the method. (**A**) Outline of the experimental procedure. Left, the transposon (green) can insert either into non-essential DNA (blue) and give rise to a clone, or into essential DNA (orange), in which case no clone is formed. Right, procedure to identify transposon insertion sites by deep-sequencing. (**B**) Profile of the transposon density across the whole genome, when the transposon original location is either a centromeric

*Figure 1 continued*

plasmid (top) or the endogenous *ADE2* locus on chromosome XV (bottom). The dashed lines indicate the chromosome centromeres. (**C**) Six examples of genomic regions and their corresponding transposon coverage in seven independent transposon libraries of indicated genotypes. Each vertical grey line represents one transposon insertion event. Genes annotated as essential are shown in orange, others in blue. Green arrowheads indicate the places where the absence of transposon coverage coincides with an essential gene. (**D**) Histogram of the number of transposons found in every annotated gene (CDS). The vertical dashed line is the median of the distribution. (**E**) Same as D, with genes categorized as non-essential (blue) and essential (orange) according to previous annotations.

The following figure supplements are available for figure 1:

**Figure supplement 1.** Size distribution of the colonies appearing on SD +Galactose -Ade.

**Figure supplement 2.** Genome-wide analysis of transposon insertion sites.

**Figure supplement 3.** Transposon density in essential and non-essential genes.

Cells with repaired *ADE2* will be able to form colonies. The excised transposon then re-inserts at random genomic locations with a frequency of ~60% (*Lazarow et al., 2012*).

We have generated seven libraries, displayed together in all figures to illustrate the reproducibility of the approach. All libraries were generated in *ade2Δ* strains derived from BY4741 and BY4742 backgrounds. Additional mutations (*dpl1Δ*, *dpl1Δ psd2Δ*, *VPS13(D716H)*, *mmm1Δ VPS13(D716H)*, *YEN1$^{on}$*, *Table 1*) will be described in the following sections. The complete dataset is available (*Supplementary file 1*) and searchable here: http://genome-euro.ucsc.edu/cgi-bin/hgTracks?hgS_doOtherUser=submit&hgS_otherUserName=benjou&hgS_otherUserSessionName=23bDePuYrk

## Detection of transposon insertion sites

Typically 7,000–10,000 colonies with a narrow size distribution (*Figure 1—figure supplement 1*) can be generated on a 8.4 cm-Ø petri dish. In the case of wild-type library 1, 240 plates yielded ~1.6E6 clones (*Table 1*). To detect transposon insertion sites, transposed cells were scraped off the 240 plates and pooled (*Figure 1A*). This pool was used to reinocculate SD medium lacking adenine (SD +Dextrose -Adenine), and the culture was grown to saturation. This step was used to dilute non-

**Table 1.** Characteristics of the libraries

| Library | Number of colonies | Reads mapped | Transposons mapped | Median read per transposon | Number of MiSeq runs | Overlap between MiSeq runs |
|---|---|---|---|---|---|---|
| Wild-type 1 | ~1.6×10$^6$ | 31794831 | 284162 | 22 | 2[*] | 54%, 88%[*] |
| Wild-type 2 | ~2.4×10$^6$ | 15303285 | 258568 | 12 | 1 | NA |
| *VPS13(D716H)* | ~4.7×10$^6$ | 24958456 | 414114 | 13 | 2[†] | 41%, 42%[†] |
| *Mmm1Δ VPS13(D716H)* | ~1.9×10$^6$ | 17799948 | 303323 | 12 | 1 | NA |
| *dpl1Δ* | ~2.3×10$^6$ | 15077156 | 401126 | 8 | 1 | NA |
| *dpl1Δ psd2Δ* | ~2.9×10$^6$ | 11649561 | 363179 | 9 | 1 | NA |
| *YEN1$^{on}$* | ~2.8×10$^6$ | 9517877 | 495125 | 6 | 1 | NA |
| Wild-type 2 + rapamycin | ~2.4×10$^6$ | 9664956 | 169322 | 9 | 1 | NA |

[*] The harvested library was grown in two flasks, one at 30°C and the other at 37°C. DNA was extracted separately from the two cultures and sequenced in two separate MiSeq runs

[†] The library was harvested as ten subpools, which were grown in ten separate flasks. DNA was extracted separately. In one case, DNA from all ten subpools was pooled and processed to sequencing in one MiSeq run. In the other case, DNAs were kept separate and processed until the PCR step (1 × 100 µl PCR by subpool). PCR products were pooled and sequenced as another MiSeq run.

transposed *ade-* cells still present on the petri dishes. The culture was then harvested by centrifugation. Genomic DNA was extracted and digested with frequent-cutting restriction enzymes, followed by ligase-mediated intramolecular circularization. Circular DNA was PCR-amplified using transposon-specific outwards-facing primers. PCR products were then sequenced on a MiSeq machine (*Figure 1A*).

## Analysis of transposon insertion sites

We aligned the sequencing reads of the wild-type library (*Table 1*) to the reference yeast genome and counted the number of mapped transposons. To account for the fact that Illumina sequencing is imperfectly accurate, we considered that two reads of the same orientation mapping within two bp of each other originated from the same transposon (see *Source code 1*). In this analysis, 284,162 independent transposons could be mapped onto the genome, representing an average density of one transposon every 42 bp, and a median number of 22 reads per transposon. No large area of the genome was devoid of transposon (*Figure 1B*). Consistent with analyses in Maize (*Vollbrecht et al., 2010*), no strong sequence preference was detected in the insertion sites (*Figure 1—figure supplement 2A*).

In many instances, though, insertion frequency was modulated along the genome with a periodicity of ~160 bp. Superimposing nucleosome occupancy data (*Lee et al., 2007*) showed that this was due to favored transposon insertion in inter-nucleosomal DNA (*Figure 1—figure supplement 2B*, *Gangadharan et al., 2010*). This effect can be appreciated at the genome-scale. Indeed, an autocorrelation analysis unraveled a ~160 bp periodic signal in the genome-wide transposon density (measured using a 40 bp moving average, *Figure 1—figure supplement 2C*). This periodicity was comparable to that of the nucleosomal density data (*Lee et al., 2007*). Finally, while no large region was devoid of transposon, some regions were actually preferentially targeted by transposons. These were the pericentromeric regions (*Figure 1B*, top), which were specifically enriched by ~20% of the transposon insertions (*Figure 1—figure supplement 2D*). The explanation for this phenomenon may pertain to nuclear architecture and to the propensity of our transposon to insert close to its excision site (*Lazarow et al., 2012*). Because the transposon is initially excised from a centromeric plasmid, and because centromeres cluster in the nuclear periphery (*Jin et al., 2000*), the transposon might tend to reinsert in the pericentromeric regions of other chromosomes. We confirmed this assumption by sequencing a small-scale library (~30 000 insertions mapped) in which the MiniDS transposon was originally at the endogenous *ADE2* locus, rather than on a centromeric plasmid. In this library, preferential targeting was not observed at pericentromeric regions, but rather in the vicinity of *ADE2*, confirming our assumption (*Figure 1B*, bottom).

## Identification of essential genes

The transposon map clearly showed that a fraction of the coding DNA sequences (CDS) were devoid of insertions. These coincided almost exactly with genes annotated as essential (*Figure 1C*, green arrowheads). The median number of transposons inserted in the CDSs of all genes was 18 per gene (*Figure 1D*). This number raised to 21 for annotated non-essential genes, but dropped to three for annotated essential genes (*Figure 1E*). This decrease was not due to a difference in the length between essential and non-essential genes, since normalizing the number of transposon insertions to the CDS length (transposon density) did not abrogate this effect (*Figure 1—figure supplement 3*). Thus our method distinguishes, in a single step, genes that are required for growth from those that are not.

Several genes, although annotated as non-essential, harbored no or very few transposons (*Supplementary file 2*). This can be attributed to the following reasons. (1) Because sequencing reads mapping to repeated sequences were discarded during alignment, repeated genes appear as transposon-free. (2) Several annotated dubious ORFs overlap with essential genes, and thus appear as transposon-free. (3) Several genes are necessary for growth in particular conditions, and are therefore not annotated as essential, yet are required in our growth conditions (SD +galactose -adenine). These include genes involved, for instance, in galactose metabolism and adenine biosynthesis.

## Identification of protein domains

It came as a surprise that some genes annotated as essential were tolerant to transposon insertion. While some of these clearly corresponded to annotation inconsistencies, many reflected an unanticipated outcome of our approach. We observed many instances of 'essential' CDSs containing transposon-permissive regions. A striking example is *GAL10* (*Figure 2A*). *GAL10* encodes a bifunctional enzyme with an N-terminal epimerase, and a C-terminal mutarotase domain (*Majumdar et al., 2004*). While the epimerase activity is indispensable in our conditions, the mutarotase activity is dispensable, as cells were fed a mixture of α- and β-D-Galactose, thus not requiring the conversion of one into the other. In accordance, the 3' end of *GAL10* was permissive for transposon insertion. The junction between the permissive and non-permissive domains of *GAL10* corresponds exactly to the junction of the two domains in the Gal10 protein. Several examples of essential genes with dispensable 3'-ends are shown in *Figure 2A*. We confirmed the dispensability of the 3' end for two genes, *TAF3* and *PRP45* (*Figure 3*).

*TAF3* encodes a 47 kDa central component of TFIID (*Sanders et al., 2002*). Our data show that only the first 76 amino acids are necessary for growth. Using homologous recombination, we replaced all but the sequence coding for the first 90 amino acids with an HA tag and a G418-resistance cassette (*KanMX6*) in a diploid strain. Sporulation and progeny segregation confirmed that strains expressing only the first 90 amino acids of Taf3 were viable (*Figure 3A*). By contrast, complete replacement of *TAF3* gave rise to only two viable, G418-sensitive spores per meiosis, confirming that *TAF3* is essential (*Figure 3B*). The essential region of Taf3 corresponds to a predicted histone-like bromo-associated domain (*Doerks et al., 2002*).

*PRP45* encodes a central component of the spliceosome. From our data, only the first 140 amino-acids of Prp45 are necessary for growth, which we confirmed using the same strategy as for *TAF3* (*Figure 3C–D*). Prp45 is predicted to be intrinsically disordered, thus no clear domain boundaries are available to rationalize our data. However, a recent cryo-EM structure of the *S. cerevisiae* spliceosome offers clues on the structure of Prp45 (*Wan et al., 2016*). Prp45 is centrally located within the spliceosome, has an extended conformation and makes several contacts with various proteins and snRNAs. In particular, the most conserved region of Prp45 is a loop that makes extensive contacts with U2 and U6 snRNAs close to the active site (*Figure 3E–F*, yellow). This loop belongs to the dispensable part of Prp45, surprisingly suggesting that splicing can occur without it. Our method thus offers insights that neither sequence conservation nor structural analysis could have predicted.

We also observed CDSs in which the 5' end is permissive for transposon insertion while the 3' end is not (*Figure 2B–C*). Again, our data show a general good coincidence between indispensable regions and annotated domains. It is surprising that transposons can be tolerated in the 5' of such genes, since several stop codons in all frames of the transposon should interrupt translation and prevent the production of the essential C-terminus. We speculate that the production of the C-terminus is enabled by spurious transcription events. A remarkable example is *SEC9* (*Figure 2C*). *SEC9* encodes a SNARE protein. The essential SNARE domain, located at the C-terminus (*Brennwald et al., 1994*), is devoid of transposons. The N-terminus of the protein is known to be dispensable for growth (*Brennwald et al., 1994*). We indeed observe several transposons inserted upstream of the SNARE domain. The extreme 5' of the gene constitutes another transposon-free region even though it encodes a dispensable part of the protein (*Brennwald et al., 1994*). It is possible that transposon insertion in this 5' region generates an unexpressed, unstable or toxic protein. Other examples of genes showing various combinations of essential domains are shown in *Figure 2C*.

We devised an algorithm to score genes according to their likelihood of bearing transposon-tolerant and -intolerant regions (*Source code 2*). In short, we computed for each CDS the longest interval between five adjacent transposons, multiplied it by the total number of transposons mapping in this CDS, and divided the result by the CDS length. We additionally imposed the following criteria: the interval must be at least 300 bp, must represent more than 10% and less than 90% of the CDS length, and a minimum of 20 transposons must map anywhere within the CDS. ~1200 genes fulfilled these requirements (*Figure 2—figure supplement 1*), of which the 400 best-scoring ones showed clear domain boundaries (*Figure 2—figure supplements 2–5*). Essential subdomains are only expected in essential genes and indeed, this gene set was overwhelmingly enriched for previously-annotated essential genes (*Figure 2—figure supplement 1*).

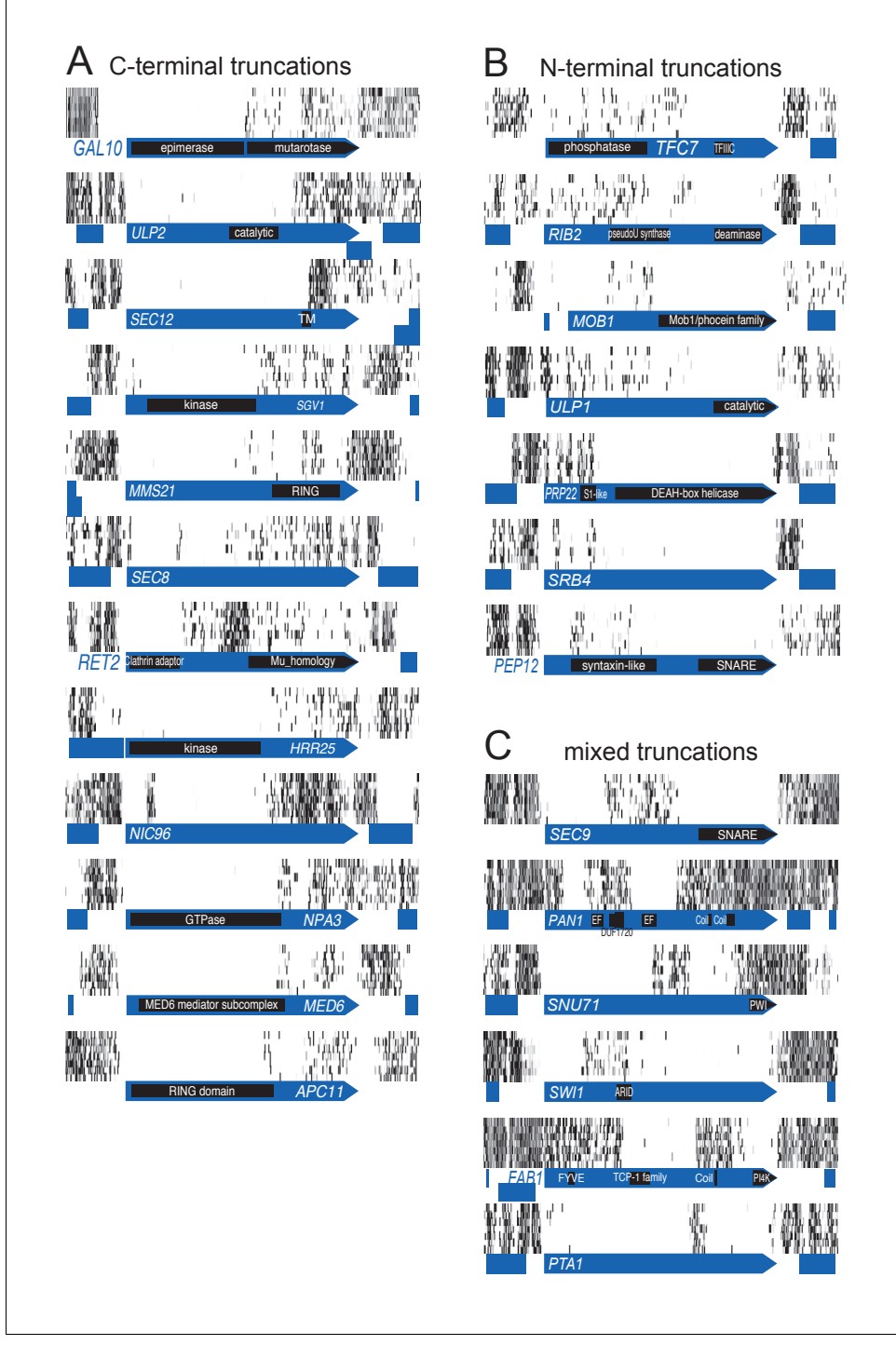

**Figure 2.** Examples of genes showing partial loss of transposon coverage. The grey level is proportional to the number of sequencing reads. Known functional domains are indicated. (**A**) Essential genes for which C-terminal truncations yield a viable phenotype. (**B**) Essential genes for which N-terminal truncations yield a viable phenotype. (**C**) Essential genes for which various truncations yield a viable phenotype.

The following figure supplements are available for figure 2:

**Figure supplement 1.** Detection of essential protein domains.

**Figure supplement 2.** Transposon maps in the 100 highest scoring genes.

*Figure 2 continued on next page*

*Figure 2 continued*

**Figure supplement 3.** Transposon maps in the genes scoring 101 to 200.

**Figure supplement 4.** Transposon maps in the genes scoring 201 to 300.

**Figure supplement 5.** Transposon maps in the genes scoring 301 to 400.

Thus, our method allows to identify not only genes, but also subdomains that are important for growth, yielding valuable structure-function information about their cognate proteins.

## Comparison of independent libraries reveals genetic interactions

Our approach can easily identify essential genes and essential protein domains. In addition, its ease makes it a potential tool to uncover genes that are not essential in standard conditions but become

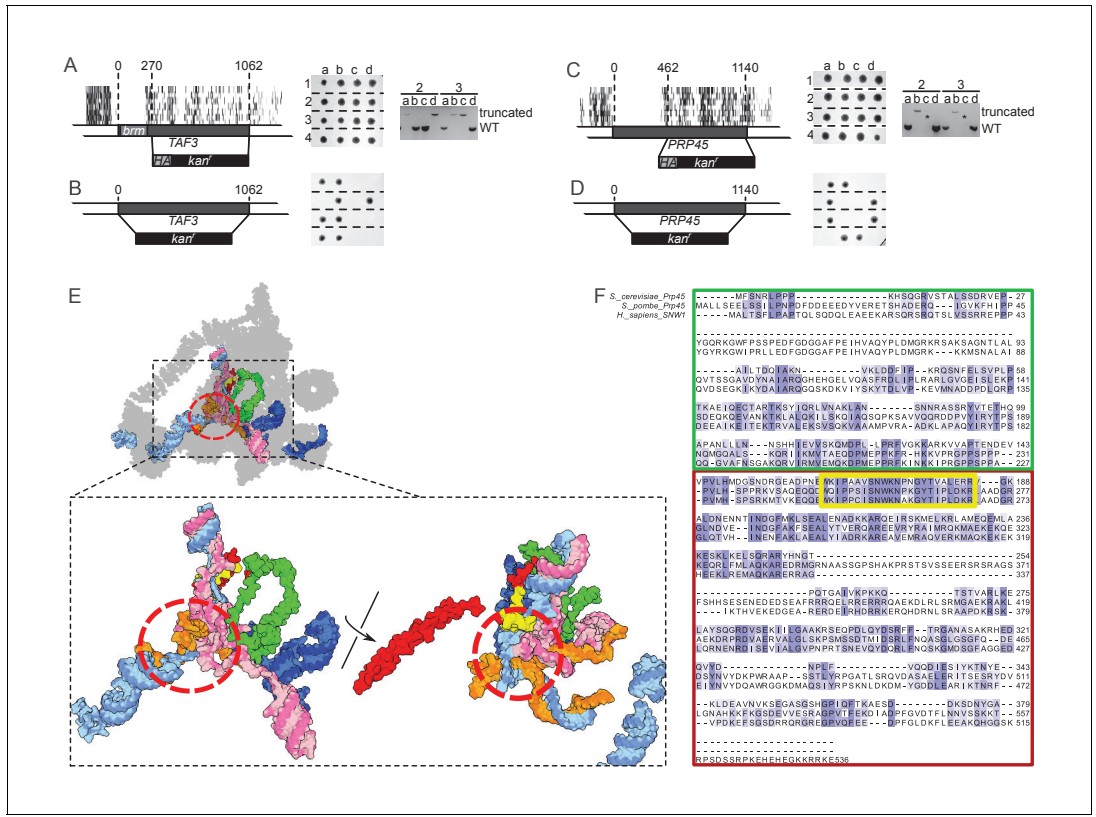

**Figure 3.** *TAF3* and *PRP45* can be truncated without visible effects on cell growth. (**A**) A truncation of *TAF3* was generated in a heterozygous diploid strain (left) by introduction of an HA tag and a G418-resistance cassette (HA kan^r). The strain was tetrad dissected (middle). Tetrads 2 and 3 were further analyzed by PCR to confirm the Mendelian segregation of the truncated allele (right). (**B**) A complete *TAF3* deletion was generated in a heterozygous diploid strain (left) by introduction of a G418-resistance cassette (kan^r). Meiosis yields only two viable, G418-sensitive spores per tetrad, confirming that *TAF3* complete deletion is lethal. (**C–D**) As in (**A–B**) but applied to *PRP45*. Asterisks in the right panel designate PCR reactions that were inefficient at amplifying the large truncated allele. The genotype of these spores can nevertheless be inferred from the Mendelian segregation of the G418 resistance. (**E**) Top, cryo-EM structure of the *S. cerevisiae* spliceosome (PDB accession 5GMK, *Wan et al., 2016*). Bottom, the same structure stripped of every protein except Prp45. The essential portion of Prp45 as defined in (**C**) is in green and the non-essential part is in red and yellow. U2, U6, U5 and substrate RNAs are depicted in pale blue, pink, dark blue and orange, respectively. The red circle indicates the catalytic active site of the spliceosome. (**F**) Alignment of the Human, *S. cerevisiae*, and *S. pombe* Prp45 orthologs. The green, red and yellow boxes are colored as in (**E**). The yellow box features the most conserved region of the protein.

important in specific conditions. Indeed, our approach yields two measures — the number of transposons mapping to a given gene, and the corresponding numbers of sequencing reads. Since in most cases, transposon insertion obliterates the gene function, both measures may be used as a proxy for fitness. We assessed the usefulness of these metrics in various genetic screens.

Synthetic genetic interaction screening is an extremely powerful approach to establish networks of functional connections between genes and biological pathways, and to discover new protein complexes (*Costanzo et al., 2010*; *Schuldiner et al., 2005*). We have recently identified single amino-acid substitutions in the endosomal protein Vps13 that suppress the growth defect of mutants of the ER-mitochondria encounter structure (ERMES) (*Lang et al., 2015b*). Suppression is dependent on the proper function of the mitochondrial protein Mcp1 (*Tan et al., 2013*) and on the endosomal protein Vam6/Vps39 (*Elbaz-Alon et al., 2014*; *Hönscher et al., 2014*). We generated a transposon library from a strain bearing both the *VPS13* suppressor allele *VPS13(D716H)* and a deletion of the ERMES component *MMM1*. In these conditions, we expected *VPS13*, *MCP1* and *VAM6/VPS39* to become indispensable, while ERMES components (*MDM10*, *MDM12* and *MDM34*) should become dispensable.

*Figure 4A* shows, for each of the 6603 yeast CDSs, the number of transposon insertion sites mapped in the wild-type (x-axis) and in the *mmm1Δ VPS13(D716H)* library (y-axis). Most CDSs fall on a diagonal, meaning that they were equally transposon-tolerant in both libraries. Consistent with the ERMES suppressor phenotype of the *VPS13(D716H)* mutation, ERMES components fell above the diagonal (that is, they bore more transposons in the *mmm1Δ VPS13(D716H)* than in the wild-type library, *Figure 4A–B*). By contrast, *VPS13*, *MCP1* and *VAM6/VPS39* fell under the diagonal, as expected (*Lang et al., 2015a*, *Figure 4, C*). Many other genes known to display synthetic sick or lethal interaction with *mmm1Δ* (*Hoppins et al., 2011*) were also found, including *TOM70*, *VPS41*, *YPT7*, *VMS1* and *YME1* (*Figure 4, C*).

As a second proof-of-principle, we generated a library from a strain in which the dihydrosphingosine phosphate lyase gene *DPL1* (*Saba et al., 1997*) was deleted, and another library from a strain in which both *DPL1* and the phosphatidylserine decarboxylase 2 gene *PSD2* (*Trotter and Voelker, 1995*) were deleted (*Figure 4D,F*). In this latter double-deleted strain, phosphatidylethanolamine can only be generated via the mitochondrial phosphatidylserine decarboxylase Psd1, and thus any gene required for lipid shuttling to and from mitochondria should become indispensable (*Birner et al., 2001*).

*LCB3*, which displays synthetic sick interaction with *DPL1* (*Zhang et al., 2001*), was less transposon-tolerant in both the *dpl1Δ* and the *dpl1Δ psd2Δ* libraries (*Figure 4D*). By contrast *PSD1*, which displays a synthetic lethality with *PSD2* on media lacking choline and ethanolamine (*Birner et al., 2001*), was transposon-intolerant in the *dpl1Δ psd2Δ* library only (*Figure 4F–G*). Interestingly, we also found that *VPS13* was less transposon-tolerant in the *dpl1Δ psd2Δ*, consistent with a role for Vps13 in interorganelle lipid exchange (*AhYoung et al., 2015*; *Kornmann et al., 2009*; *Lang et al., 2015a*, *2015b*).

Additionally, when comparing the *dpl1Δ* and wild-type libraries, one of the best hits was the histidine permease *HIP1* (*Figure 4D–E*). This did not, however, reflect a genetic interaction between *HIP1* and *DPL1*, but instead between *HIP1* and *HIS3*; during the construction of the strains, *ADE2* was replaced by a *HIS3* cassette in the wild-type strain, while it was replaced by a NAT cassette in the *dpl1Δ* strain. The histidine-auxotroph *dpl1Δ* strain, therefore required the Hip1 permease to import histidine.

Thus, synthetic genetic interactions are visible through pairwise comparison of the number of transposons per genes. However, this metrics leads to a significant spread of the diagonal (*Figure 4A,D,F*). This spread is due to the intrinsic noise of the experiment. Indeed, the number of transposons per gene is expected, for each gene, to follow a binomial distribution. Sampling variability may thus mask biologically relevant differences. To overcome this limitation, we reasoned that comparing sets of one or more libraries against each other, rather than comparing two libraries in a pairwise fashion (as in *Figure 4*), would greatly improve the signal-to-noise ratio. We thus calculated an average fold-change of the number of transposons per gene between the experimental and reference sets, as well as a *p*-value (based on a Student's t-test) associated with this change. The fold-change and *p*-values were then plotted as a volcano plot (*Figure 4—figure supplement 1*, *Supplementary file 3*). In volcano plots, synthetic genes appeared well separated from the bulk of

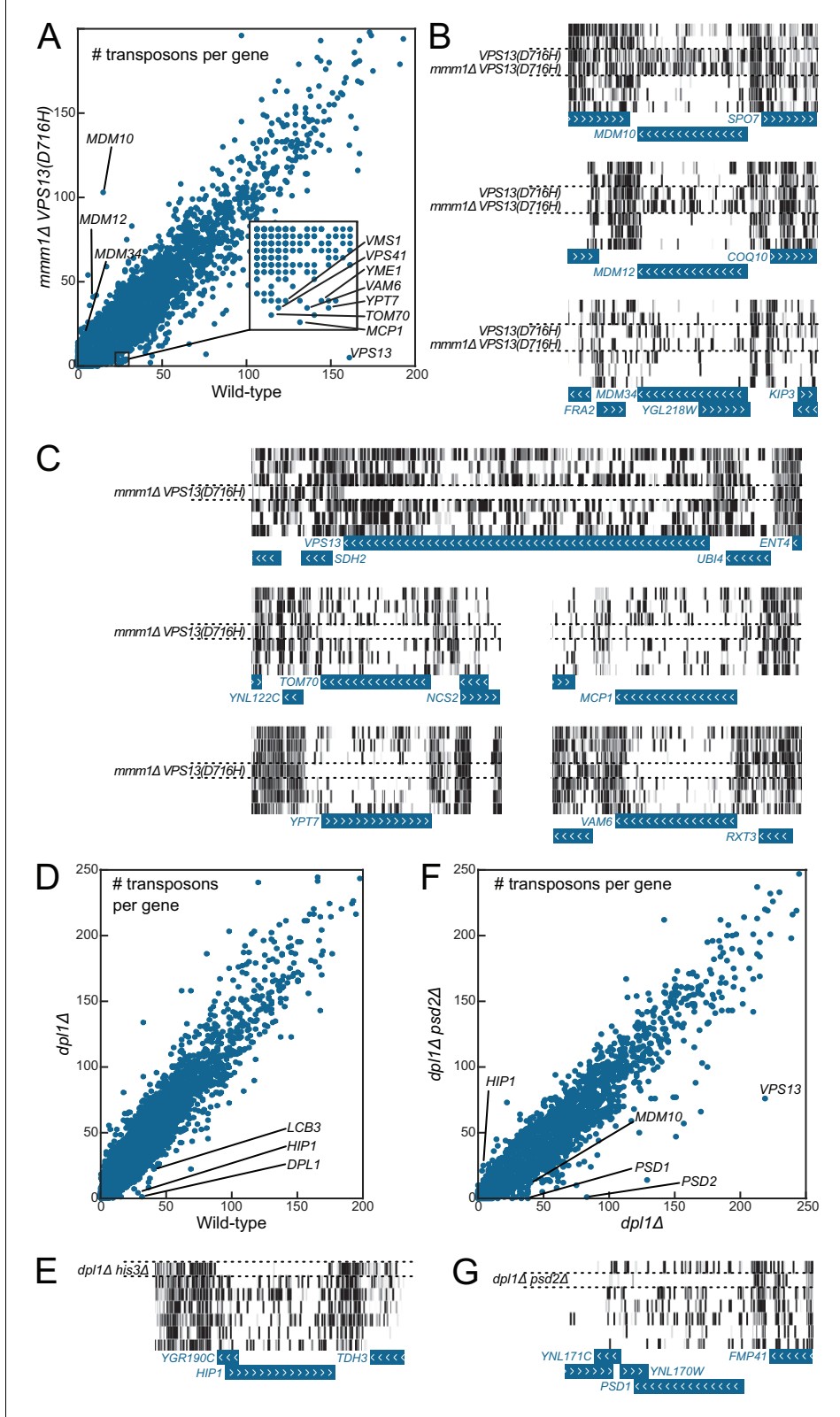

**Figure 4.** Genetic interaction analyses. Libraries in panels B, C, E and G are displayed in the same order as in *Figure 1C*. (**A**) Comparison of the number of transposons inserted in each of the 6603 yeast CDSs in the wild-type (x-axis) and *mmm1Δ VPS13(D716H)* (y-axis) libraries. (**B**) Transposon coverage of genes encoding ERMES components is increased in libraries from strains bearing the *VPS13(D716H)* allele. (**C**) Examples of genes showing

*Figure 4 continued on next page*

*Figure 4 continued*

synthetic sick/lethal interaction with *mmm1Δ VPS13(D716H)*. (D) Comparison of the number of transposons inserted in each of the 6603 yeast CDSs in the wild-type (x-axis) and *dpl1Δ* (y-axis) libraries. (E) Transposon coverage of the *HIP1* locus in the *dpl1Δ his3Δ* library and in all the other libraries (*HIS3*). (F) Comparison of the number of transposons inserted in each of the 6603 yeast CDSs in the *dpl1Δ* (x-axis) and *dpl1Δ psd2Δ* (y-axis) libraries. (G) Transposon coverage of the *PSD1* locus in the *dpl1Δ psd2Δ* and in all other libraries.

The following figure supplement is available for figure 4:

**Figure supplement 1.** Volcano plots comparing libraries or combinations of libraries as indicated.

neutral genes, showing that parallel library comparison is a robust way to increase the signal-to-noise ratio.

Synthetic lethality is one type of genetic interaction. Another type is rescue of lethal phenotype, where a gene deletion is lethal in an otherwise wild-type strain but can be rescued by a suppressor mutation. We describe two such phenomena observed in our libraries. The first concerns the septin gene *CDC10*. Septin proteins are cytoskeletal GTPases that form a ring at the bud neck. This structure is necessary for vegetative growth in *S. cerevisiae*, and all septin genes are essential with the exception of *CDC10* (*Bertin et al., 2008*; *McMurray et al., 2011*). Indeed, at low temperature, *cdc10Δ* cells are viable and able to assemble a septin ring. This Cdc10-less ring is based on a Cdc3-Cdc3 interaction, instead of the normal Cdc3-Cdc10 interaction (*McMurray et al., 2011*). Because the propensity of Cdc3 to self-assemble is weak, low temperature is thought to be necessary to stabilize the interaction. Since we grew all libraries at moderately high temperature (30°C), *CDC10* was, as expected, essentially transposon-free in most libraries (*Figure 5A*). In the *dpl1Δ psd2Δ* library, however, the number of transposons mapping within *CDC10* increased significantly, indicating that the absence of Psd2 and Dpl1 suppressed the *cdc10Δ* phenotype (*Figure 5A*, *Figure 4—figure supplement 1*, bottom left). Genetic analysis revealed that the *dpl1Δ* allele alone allowed *cdc10Δ* cells to grow at higher temperature, indicating that the Cdc10-less septin ring was stabilized in the absence of Dpl1 (*Figure 5B–C*). Genetic analysis also demonstrated that the rescue of *cdc10Δ* by *dpl1Δ* was independent of *PSD2*. It is unclear why the suppressive effect was detected in the *dpl1Δ psd2Δ* library, but not in the *dpl1Δ* library. We speculate that differences in growth conditions between experiments have obscured either the suppression in the *dpl1Δ* library or the involvement of Psd2 in our tetrad analyses.

Dpl1 is an ER protein that does not contact any of the septin subunits; its destabilizing effect on the septin ring must therefore be indirect. Since Dpl1 is a regulator of sphingolipid precursors (*Saba et al., 1997*) and since the septin ring assembles in contact with the plasma membrane (*Bertin et al., 2010*), it is most likely the changing properties of the membrane in *DPL1*-deficient cells that allow or restrict the assembly of Cdc10-less septin rings. This hypothesis is particularly appealing because temperature has a profound effect on membrane fluidity and composition (*Ernst et al., 2016*). Thus, the stabilizing effect of low temperature on Cdc10-less septin rings might not only be the result of a direct stabilization of Cdc3-Cdc3 interaction, but also of changes in plasma membrane properties, which can be mimicked by *DPL1* ablation.

The second example of rescue of a lethal phenotype was observed in a library made from a strain expressing a constitutively active version of the Holliday-junction resolvase Yen1, a member of the Xeroderma Pigmentosum G (XPG) family of 5'-flap endonucleases (*Ip et al., 2008*). In wild-type strains, Yen1 is kept inactive during the S-phase of the cell cycle via Cdk-mediated phosphorylation (*Matos et al., 2011*). Rapid dephosphorylation at anaphase activates Yen1 for the last stages of mitosis. When nine Cdk1 target sites are mutated to alanine, Yen1 becomes refractory to inhibitory phosphorylation and active during S-phase (Yen1$^{on}$) (*Blanco et al., 2014*). To investigate the cellular consequence of a constitutively active Yen1, we generated a library in a *YEN1$^{on}$* background. We discovered that, under these conditions, the essential gene *DNA2* became tolerant to transposon insertion (*Figure 5D*). Further genetic analyses confirmed that the presence of the *YEN1$^{on}$* allele led to a rescue of the lethal phenotype of *dna2Δ* strains; spores bearing the *dna2* deletion failed to grow unless they additionally bore the *YEN1$^{on}$* allele (*Figure 5E*). Moreover, at 25°C, the colony size of *dna2Δ YEN1$^{on}$* spores was comparable to that of the *DNA2* counterparts (*Figure 5E*, right).

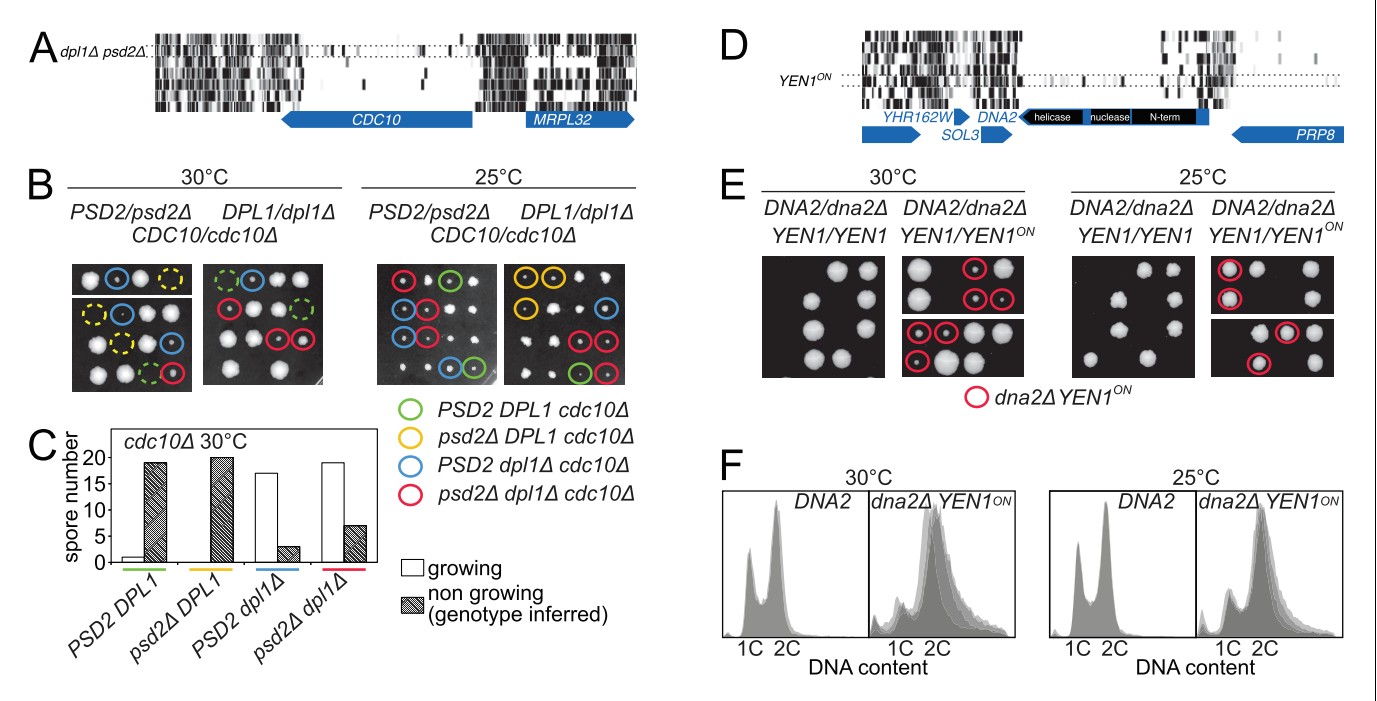

**Figure 5.** Synthetic rescue of lethal phenotypes. (**A**) Transposon coverage of *CDC10* in the seven libraries. The coverage is increased in the *dpl1Δ psd2Δ* library. (**B**) Tetrad dissection of a *PSD2/psd2Δ DPL1/dpl1Δ CDC10/cdc10Δ* triple heterozygote at 30°C (left) and 25°C (right). The *cdc10Δ* spores of ascertained genotype are circled with a color-coded solid line. *cdc10Δ* spores for which the genotype can be inferred from the other spores of the tetrad are circled with a color-coded dashed line. (**C**) Quantification of growing and non-growing *cdc10Δ* spores of the indicated genotype obtained from 48 tetrads (three independent diploids). (**D**) Transposon coverage of *DNA2* in the seven libraries. The coverage is increased in the *YEN1^{on}* library. (**E**) Tetrad dissection of a *DNA2/dna2Δ YEN1/YEN1* single heterozygote and of a *DNA2/dna2Δ YEN1/YEN1^{on}* double heterozygote at 30°C (left) and 25°C (right). All viable *dna2Δ* spores additionally carry the *YEN1^{on}* allele (red circle). (**F**) FACS profile of propidium-iodide-stained DNA content in *DNA2* and *dna2Δ YEN1^{on}* strains exponentially growing at 30°C (left) and 25°C (right). For *DNA2* panels, each profile is an overlay of two independent strains. For *dna2Δ YEN1^{on}* panels, each profile is an overlay of four independent strains.

However, FACS analysis of DNA content revealed that *dna2Δ YEN1^{on}* cells accumulated in S- and G2-phases (*Figure 5F*).

*DNA2* encodes a large protein with both helicase and endonuclease activities (*Budd and Campbell, 1995*). Interestingly, while the helicase activity can be disrupted without affecting yeast viability, the nuclease activity is essential (*Ölmezer et al., 2016*), presumably due to its involvement in processing long 5'-flaps of Okazaki fragments. Our genetic data now reveal that Yen1 is able to partially fulfill the essential roles of Dna2, provided that its activity is unrestrained in S-phase. Since the spectrum of Yen1 substrates includes 5'-flap structures (*Ip et al., 2008*), Yen1^{on} may be able to substitute the essential function of Dna2 by providing 5'-flap nuclease activity in S-phase. This finding extends previous work showing that Yen1 serves as a backup for the resolution of replication intermediates that arise in helicase-defective mutants of Dna2 (*Ölmezer et al., 2016*).

Thus, our method can be used to screen for negative and positive genetic interactions in a rapid, labor-efficient and genome-wide manner, in strains bearing single and multiple mutations.

## Chemical genetics approach

To assess our method's ability to uncover drug targets, we used the well-characterized immune-suppressive macrolide rapamycin. Rapamycin blocks cell proliferation by inhibiting the target of rapamycin complex I (TORC1), through binding to the Fk506-sensitive Proline Rotamase Fpr1 (*Heitman et al., 1991*). The TORC1 complex integrates nutrient-sensing pathways and regulates cellular growth accordingly. Rapamycin treatment therefore elicits a starvation response, which stops proliferation. We generated and harvested a wild-type library, then grew it in medium containing

rapamycin at low concentration. To compare it to an untreated wild-type library, we counted the number of sequencing reads mapping to each of the 6603 yeast CDSs in both conditions (*Figure 6A*). Most genes fell on a diagonal, as they do not influence growth on rapamycin. By contrast, a few genes were robustly covered by sequencing reads in the rapamycin-treated library, indicating that their interruption conferred rapamycin resistance. Expectedly, the best hit was *FPR1*, encoding the receptor for rapamycin (*Hall, 1996*). Other hits included *RRD1* (Rapamycin-Resistant Deletion 1), *TIP41*, *GLN3*, *SAP190*, *PSP2*, *CCS1*, *ESL2* and members of the PP2A phosphatase *PPH21* and *PPM1* (*Figure 6A*). These genes are either directly involved in rapamycin signaling or known to confer rapamycin resistance upon deletion (*Xie et al., 2005*).

Finding the TORC1 regulator *PIB2* (*Kim and Cunningham, 2015*) was however unexpected, because *PIB2* deletion confers sensitivity, not resistance, to rapamycin (*Kim and Cunningham, 2015*; *Parsons et al., 2004*). To solve this conundrum, we looked at the distribution of transposons on the *PIB2* coding sequence. All the insertions selected by rapamycin treatment mapped to the 5'-end of the gene (*Figure 6B*). On the contrary, the rest of *PIB2* was less covered in the rapamycin-treated than in non-rapamycin-treated libraries (*Figure 6—figure supplement 1A*). Insertions in the 5' end of *PIB2* therefore conferred rapamycin resistance, while insertions elsewhere, like complete deletion of *PIB2*, conferred rapamycin sensitivity.

To confirm this result, we engineered N-terminal truncations of Pib2, guided by the transposon map. Strains expressing Pib2 variants that were truncated from the first until up to amino acid 304 were hyperresistant to rapamycin (*Figure 6C*, top). We also confirmed that, by contrast, complete *PIB2* deletion caused rapamycin hypersensitivity. A larger N-terminal truncation extending beyond the mapped territory of rapamycin resistance-conferring transposons, did not mediate hyperresistance to rapamycin (*Figure 6G*, Pib2$^{426-635}$).

To assess whether rapamycin-hyperresistant *PIB2* truncations behaved as gain-of-function alleles, we co-expressed Pib2$^{165-635}$ and full-length Pib2. In these conditions, the rapamycin hyperresistance was mitigated, indicating that the effect of the truncated Pib2 protein was semi-dominant (*Figure 6C*, bottom). Expressing the truncation from a high-copy (2 μ) vector did not further increase resistance to rapamycin, indicating that higher expression levels did not change the semi-dominance of the *PIB2* truncation allele.

Thus, Pib2 truncation leads to a gain-of-function that translates into semi-dominant rapamycin hyperresistance. Because gain-of-function alleles of *PIB2* lead to rapamycin resistance, while loss-of-function alleles lead to sensitivity, our data suggest that Pib2 positively regulates TORC1 function. To test this idea further, we investigated the effects of full-length and various truncations of Pib2 on TORC1 signaling. Switching yeast cells from a poor nitrogen source (proline) to a rich one (glutamine) triggers a fast activation of TORC1 (*Stracka et al., 2014*), leading to a transient surge in the phosphorylation of Sch9, a key target of TORC1 in yeast (*Urban et al., 2007*). We cultured cells on proline-containing medium and studied TORC1 activation 2 min following addition of 3 mM glutamine by determining the phosphorylation levels of Sch9-Thr$^{737}$. Pib2 deficiency severely blunted TORC1 activation in these conditions (*Figure 6D*, top). In a control experiment, a strain lacking *GTR1* – a component of the TORC1-activating EGO (Exit from rapamycin-induced GrOwth arrest) complex, which is orthologous to the mammalian Rag GTPase-Ragulator complex (*Chantranupong et al., 2015*; *Powis and De Virgilio, 2016*) – showed a similarly blunted response with respect to TORC1 activation following glutamine addition to proline-grown cells (*Figure 6D*, top). By contrast, glutamine-mediated TORC1 activation appeared normal in a strain expressing an N-terminally truncated Pib2 variant (*pPIB2$^{165-635}$*, *Figure 6D*, bottom). Thus, like Gtr1, Pib2 is necessary to activate TORC1 in response to amino acids. The N-terminus of Pib2 appears to be an inhibitory domain. The ablation of this domain confers rapamycin resistance, yet is not sufficient to constitutively activate TORC1 (*e.g.* in proline-grown cells).

Having identified an inhibitory activity at the N-terminus of Pib2, we proceeded to address the function of other Pib2 domains. To this aim, we used a split-ubiquitin-based yeast-two-hybrid assay to probe the interaction of Pib2 fragments with the TORC1 component Kog1, and studied the rapamycin resistance of strains expressing various truncations of Pib2 (*Figure 6E–F*). We found that Pib2 harbored at least two central Kog1-binding regions, since two mostly non-overlapping fragments (Pib2$^{1-312}$ and Pib2$^{304-635}$) showed robust interaction with Kog1 (*Figure 6F* and *Figure 6—figure supplement 1B*). Kog1 binding is, however, not essential for Pib2-mediated TORC1 activation, since

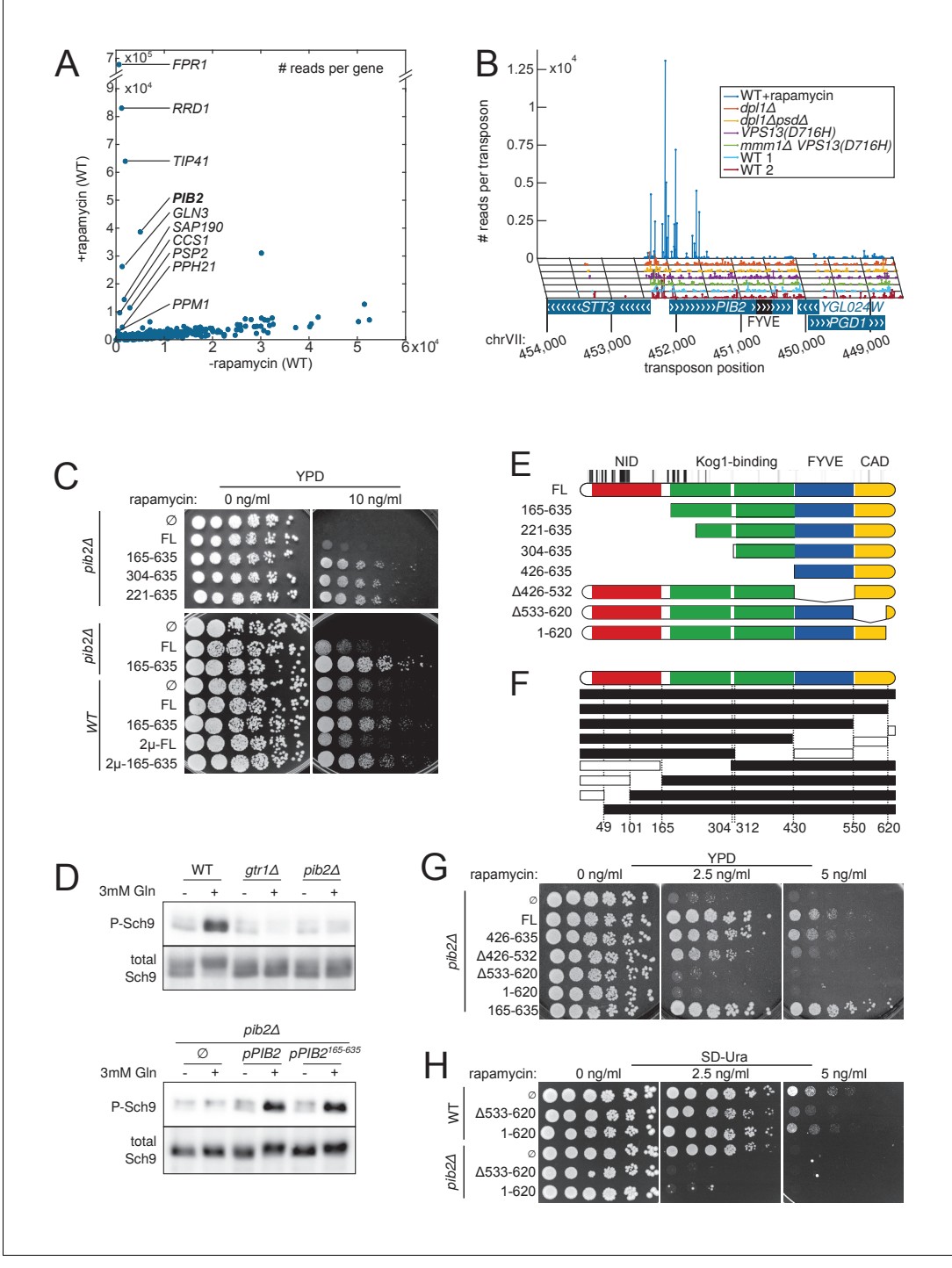

**Figure 6.** Detection of rapamycin resistant strains. (**A**) Comparison of the number of sequencing reads mapping to each of the 6603 yeast CDSs in rapamycin-untreated (x-axis) and -treated (y-axis) libraries. Note the difference in scale between both axis due to the high representation of rapamycin-resistant clones. (**B**) Distribution of transposons and number of associated sequencing reads on the *PIB2* gene. Transposons with high number of sequencing reads in the rapamycin-treated library are clustered at the 5'-end of the CDS. (**C**) Wild-type (WT) and *pib2Δ* strains were transformed with either an empty plasmid (∅) or plasmids encoding full-length (FL) or indicated fragments of Pib2 (see E, numbers refer to the amino acid position in the full-length Pib2 protein). 5-fold serial dilutions of these strains were spotted on YPD or YPD containing 10 ng/ml rapamycin. Centromeric plasmids were used in all strains, except in those denoted with 2 μ, which carried multi-copy plasmids. (**D**) Strains of the indicated genotypes, transformed with either an empty plasmid (∅) or plasmids encoding full length (*pPIB2*) or truncated

*Figure 6 continued on next page*

*Figure 6 continued*

(*pPIB2*[165-635]) versions of Pib2, were grown exponentially in minimal medium with proline as nitrogen source. 3 mM glutamine was added to the culture and the cells were harvested 2 min later. Protein extracts were resolved on an SDS-page and probed with antibodies either specific for Sch9-pThr[737] (P-Sch9), or for total Sch9 to assess TORC1 activity. (E) Schematic overview of Pib2 architecture and of the fragments used for genetic studies. (F) Summary of yeast-two-hybrid interactions between Pib2 fragments and the TORC1 subunit Kog1 (*Figure 6—figure supplement 1*). Fragments indicated by a black box interacted with Kog1, fragments indicated by a white box did not. (G) *pib2Δ* cells expressing the indicated Pib2 fragments from plasmids (see E) were assayed for their sensitivity to rapamycin (2.5 or 5 ng/ml) as in C. (H) WT or *pib2Δ* cells expressing the indicated Pib2 fragments from plasmids were assayed as in G, except that cells were spotted on synthetic medium to apply a selective pressure for plasmid maintenance.

The following figure supplement is available for figure 6:

**Figure supplement 1.** A) Transposon coverage of the *PIB2* gene.

cells expressing a fragment without these Kog1-binding domains (Pib2[426-635]) were as rapamycin-resistant as cells expressing full length Pib2 (FL, *Figure 6G*).

Pib2 harbors a phosphatidylinositol-3-phosphate (PI3P) -binding Fab1-YOTB-Vac1-EEA1 (FYVE) domain (*Figure 6E*). Pib2 truncations lacking the FYVE domain are unable to bind PI3P, and hence to properly localize to the vacuole (*Kim and Cunningham, 2015*). When cells expressed FYVE domain-truncated Pib2 (Pib2[Δ426-532]), their rapamycin resistance decreased, but not as severely as observed in *pib2Δ* strains (*Figure 6G*). This indicates that FYVE-domain-mediated vacuolar recruitment is not absolutely required for Pib2 to activate TORC1.

Strikingly, cells expressing Pib2 variants that were either truncated at the extreme C-terminus (*PIB2*[1-620]) or carried a deletion within the C-terminus (*PIB2*[Δ533-620]) were as sensitive to rapamycin as *pib2Δ* cells (*Figure 6G*). The C-terminus of Pib2 is therefore important to ensure proper TORC1 activation.

We conclude that Pib2 harbors the following functional domains: a large N-terminal Inhibitory Domain (NID), and a C-terminal TORC1-Activating Domain (CAD); the central portion of the protein harbors FYVE and Kog1-binding domains for proper targeting of Pib2 to the vacuole and TORC1, respectively.

Could the NID act in an auto-inhibitory allosteric fashion by preventing the CAD from activating TORC1? We reasoned that if this were the case, plasmid-encoded N-terminally truncated Pib2 should confer similar levels of rapamycin resistance independently of whether a genomic wild-type *PIB2* copy was present or not. However, wild-type cells expressing N-terminally truncated Pib2[165-635] from a centromeric or a multi copy 2 μ plasmid were less resistant to rapamycin than *pib2Δ* cells expressing Pib2[165-635] from a centromeric plasmid (*Figure 6C*). Thus, the NID activity of the endogenously-expressed wild-type Pib2 is able to mitigate the rapamycin resistance conferred by ectopic expression (or overexpression) of the CAD. Therefore, our data suggest that the NID does not auto-inhibit the CAD within Pib2, but rather, that the NID and CAD act independently and antagonistically on TORC1.

This latter scenario predicts that, just as expressing NID-truncated Pib2 semi-dominantly activates TORC1 (*Figure 6C*), expressing CAD-truncated Pib2 should semi-dominantly inhibit it. To test this prediction, we expressed the two CAD truncations Pib2[Δ533-620] and Pib2[1-620] in an otherwise wild-type strain. The resulting strains were significantly more sensitive to rapamycin than their counterparts expressing only full length Pib2 (*Figure 6H*). Furthermore, when *pib2Δ* cells expressed the Pib2 CAD truncation alleles, they became even more sensitive to rapamycin. Therefore, Pib2-NID does not act in an auto-inhibitory manner, but rather inhibits TORC1 independently of the presence of a Pib2-CAD.

Since the Rag GTPase Gtr1-Gtr2 module of the EGO complex also mediates amino acid signals to TORC1 (*Binda et al., 2009*; *Dubouloz et al., 2005*, see also *Figure 6D*), we tested the possibility that Pib2-NID inhibits TORC1 by antagonizing Gtr1-Gtr2. This does not appear to be the case; first, the expression of Pib2[Δ533-620] or Pib2[1-620] further enhanced the rapamycin sensitivity of *gtr1Δ gtr2Δ* cells (*Figure 6—figure supplement 1C*), and second, the Pib2[Δ533-620]- or Pib2[1-620]-mediated

rapamycin hypersensitivity was not suppressed by expression of the constitutively active, TORC1-activating Gtr1$^{Q65L}$-Gtr2$^{S23L}$ module (*Binda et al., 2009*; *Figure 6—figure supplement 1D*). These results suggest that Pib2-NID inhibits TORC1 independently of the EGO complex.

In summary, Pib2 is targeted to TORC1 by binding to the vacuolar membrane through its FYVE domain and to Kog1 via its middle portion. Pib2 harbors two antagonistic activities, one activating and the other repressing TORC1. The dose-independent semi-dominant nature of the respective truncations indicates that both repressing and activating activities influence TORC1 independently and do not appear to compete for the same sites on TORC1.

We speculate that high-quality amino acids, such as glutamine, balance the antagonistic TORC1-activating and -repressing activities of Pib2 to tune growth rate according to available resources. In this context, it will be interesting to elucidate how the activities of Pib2 and the EGO complex are coordinated to stimulate TORC1 in response to amino acids.

## Discussion

Here we present a novel method based on random transposon insertion and next-generation sequencing, to functionally screen the genome of *Saccharomyces cerevisiae*. SAturated Transposon Analysis in Yeast (SATAY) can reveal positive and negative genetic interactions, auxotrophies, drug-sensitive or -resistant mutants, and map functional protein domains. SATAY combines advantages from both classical genetics and high-throughput robotic screens. SATAY can in principle be implemented in any strain background, since it does not rely on the existence of available deletion libraries and does not necessitate markers to follow genetic traits. Moreover, SATAY explores a wide genetic space; exotic alleles can be generated as exemplified by *PIB2* (*Figure 6* and *Figure 6—figure supplement 1*), where transposon insertions in different regions of the gene generate opposite phenotypes. Transposon insertion is not limited to CDSs; we observe that promoters of essential genes are often transposon-intolerant (*Figure 2*, see *GAL10, SGV1, MMS21, RET2, HRR25, NPA3, SEC9, SWI1*). We also observe that known essential tRNA, snRNA, as well as SRP RNA genes are transposon intolerant (see Supplementary Dataset). Finally, our data reveal transposon-intolerant areas of the genome that do not correspond to any annotated feature, indicating that SATAY could help discover yet-unknown functional elements.

SATAY yields unprecedented insight on the domain structure-function relationship of important proteins, and allows the mapping of important functional domains, without prior knowledge. Dispensable domains in essential proteins might not be required for the essential function of these proteins, but may have other roles. The sub-gene resolution enabled by SATAY may thus unveil yet-unsuspected accessory or regulatory functions, even in otherwise well-studied proteins. In addition, structure-function information revealed by SATAY may guide 3D-structure determination efforts by indicating possibly flexible accessory domains.

The resolution of a SATAY screen is directly proportional to the number of transposons mapped onto the genome. The current resolution is ~1/40 bp, which is amply sufficient to confidently identify essential genes and protein domains. This resolution is achieved by mapping ~300,000 transposons, starting from a 1.6E6-colonies library. Not every colony generates a detectable transposon. This is due to several reasons. (1) Excised transposons only reinsert in 60% of the cases (our observations and *Lazarow et al., 2012*). (2) 7% of the sequencing reads mapped onto repetitive DNA elements (such as rDNA, Ty-elements and subtelomeric repeats) and were discarded because of ambiguous mapping. (3) Two transposons inserted in the same orientation within two bp of each other will be considered as one by our algorithm. This might be exacerbated at densely covered areas of the genome, such as pericentromeric regions. (4) Some transposon insertion products may not be readily amplifiable by our PCR approach.

We observe that increasing both the size of the original library and the sequencing depth leads to an increase in mapped transposon number (albeit non-proportional, *Table 1*). Therefore, the resolution of a screen can be tailored according to the question and the available resource.

We could not detect a preferred sequence for transposon insertion (*Figure 1—figure supplement 2A*), yet two features of the yeast genome biased insertion frequency in select regions. The first is nucleosome position; the Ds transposon has a tendency to insert more frequently within internucleosomal regions, indicating that, like other transposases (*Gangadharan et al., 2010*), Ac might have better access to naked DNA (*Figure 1—figure supplement 2B–C*). The second pertains to a

tendency of the transposon to integrate in the spatial proximity of its original location (*Lazarow et al., 2012*). Indeed, when originated from a centromeric plasmid, the transposon has an increased propensity to reinsert in the pericentromeric regions of other chromosomes. This is not due to a particular feature of pericentromeric chromatin, since this propensity is lost when the transposon original location is on the long arm of ChrXV (*Figure 1B*). Instead, the increased propensity is likely resulting from the clustering of centromeres in the nuclear space (*Jin et al., 2000*), indicating that centromeric plasmids cluster with the chromosomal centromeres. Thus, SATAY can be utilized to probe, not only the function of the genome, but also its physical and regulatory features, such as nucleosome position and 3D architecture.

Finally, SATAY does not require any particular equipment besides access to a sequencing platform, and involves a limited amount of work. It can be easily multiplexed to perform several genome-wide screens simultaneously. Each screen yields two measures, the number of transposons and the number of sequencing reads, both of which, for each gene, reveal the fitness of the cognate mutant. While the number of transposons per gene is appropriate to look for genes that become important for growth, as in genetic interaction screens, the number of sequencing reads is better suited to identify strains that are positively selected, like drug-resistant mutants. Both metrics suffer from intrinsic noise, stemming from the inherently discrete structure of the data, and probably also from unavoidable biases in the amplification and sequencing of the libraries. We show that this noise can be reduced by comparing multiple libraries against each other (*Figure 4—figure supplement 1*). Moreover, comparing multiple libraries allows to tailor the composition of each library set to needs. For instance, grouping the *VPS13(D716H)* with the *mmm1Δ VPS13(D716H)* libraries allows to selectively detect synthetic interactions with *VPS13(D716H)* (e.g. ERMES components). By contrast, comparing the *mmm1Δ VPS13(D716H)* library with all others, selectively finds genes important for the ERMES suppression phenomenon. Thus, while signal-to-noise ratio might be a limiting factor for the detection of genetic interactions, we anticipate that increasing the number of libraries, for instance by generating multiple libraries in each condition, will likely decrease the incidence of false positive and false negative. With the increasing number of screens performed in various conditions will also come the ability to find correlation patterns among genes that are required or dispensable for growth in similar sets of conditions. Such correlations are accurate predictors of common functions and have been extensively used in synthetic genetic screens, such as the E-MAP procedure (*Kornmann et al., 2009*; *Schuldiner et al., 2005*). However, while E-MAP screens compute patterns of genetic interaction on a subset of chosen genetic interaction partners, SATAY allows to detect genetic interactions at the genome scale.

Because the approach only necessitates a transposon and a transposase, it should not only be feasible in *S. cerevisiae* and *S. pombe* (*Guo et al., 2013*), but also amenable to other industrially-, medically- or scientifically-relevant haploid fungi, such as *Y. lipolytica*, *C. glabrata*, *K. lactis* and *P. pastoris*.

The Ds transposon can in principle accommodate extra sequences with no known length limitation (*Lazarow et al., 2013*). An interesting future development will be to incorporate functional units in the transposon DNA, for instance strong promoters, repressors or terminators, IRESs, recognition sites for DNA-binding proteins (LacI or TetR), recombination sites (LoxP, FRP), or coding sequences for in-frame fusion, such as GFP, protein- or membrane-binding domains, signal sequences, etc. Improved designs will not only permit finer mapping of protein domains without reliance on spurious transcription and translation, but might allow the exploration of an even wider genetic space, for instance by generating gain-of-function variants, thus enabling the development of novel approaches to interrogate the yeast genome.

## Materials and methods

### Plasmids and strains

All yeast strains, oligonucleotides and plasmids used herein are listed in *Tables 2*, *3* and *4*, respectively. To generate pBK257, the *ADE2* gene interrupted with the MiniDs transposon was PCR amplified from strain CWY1 (*Weil and Kunze, 2000*), using PCR primers #4 and #5. The PCR product and pWL80R_4x (plasmid encoding the Ac transposase under the control of the *GAL1* promoter, *Lazarow et al., 2012*) were digested with SacI, then ligated together. This plasmid does not confer

**Table 2.** Yeast strains used in this study.

| Name | Parent | Genotype | Reference |
|---|---|---|---|
| CWY1 | BY4723 | MATa his3Δ0 ura3Δ0 ade2:Ds-1 | *Weil and Kunze (2000)* |
| ByK157 | BY4743 | MATα his3Δ1 leu2Δ0 lys2Δ0 ura3Δ0 VPS13(D716H) | *Lang et al. (2015b)* |
| ByK352 | BY4741 | MATa his3Δ1 leu2Δ0 met17Δ0 ura3Δ0 ade2Δ::HIS3* | This study |
| ByK484 | By4742 | MATα his3Δ1 leu2Δ0 lys2Δ0 ura3Δ0 ade2Δ::HIS3* | This study |
| ByK485 | ByK352 and ByK484 | MATa/α his3Δ1/his3Δ1 leu2Δ0/leu2Δ0 LYS2/lys2Δ0 met17Δ0/MET17 ura3Δ0/ura3Δ0 ade2Δ::HIS3*/ ade2Δ::HIS3* | This study |
| ByK446 | ByK157 | MATα his3Δ1 leu2Δ0 ura3Δ0 ade2Δ::HIS3* VPS13(D716H) | This study |
| ByK528 | ByK446 | MATα his3Δ1 leu2Δ0 ura3Δ0 ade2Δ::HIS3* VPS13(D716H) mmm1Δ::KanMX6 | This study |
| ByK530 | ByK352 | MATa his3Δ1 leu2Δ0 met17Δ0 ura3Δ0 ade2Δ::NAT* dpl1Δ::KanMX6 | This study |
| ByK533 | ByK352 | MATa his3Δ1 leu2Δ0 met17Δ0 ura3Δ0 ade2Δ::HIS3* psd2Δ::KanMX6 dpl1Δ::NAT | This study |
| ByK576 | ByK485 | MATa/α his3Δ1/his3Δ1 leu2Δ0/leu2Δ0 LYS2/lys2Δ0 met17Δ0/MET17 ura3Δ0/ura3Δ0 ade2Δ::HIS3*/ ade2Δ::HIS3* prp45Δ::KanMX6/PRP45 | This study |
| ByK579 | ByK485 | MATa/α his3Δ1/his3Δ1 leu2Δ0/leu2Δ0 LYS2/lys2Δ0 met17Δ0/MET17 ura3Δ0/ura3Δ0 ade2Δ::HIS3*/ ade2Δ::HIS3* PRP45$^{1-462}$-HA(KanMX6)/ PRP45 | This study |
| ByK583 | ByK485 | MATa/α his3Δ1/his3Δ1 leu2Δ0/leu2Δ0 LYS2/lys2Δ0 met17Δ0/MET17 ura3Δ0/ura3Δ0 ade2Δ::HIS3*/ ade2Δ::HIS3* taf3Δ::KanMX6/TAF3 | This study |
| ByK588 | ByK485 | MATa/α his3Δ1/his3Δ1 leu2Δ0/leu2Δ0 LYS2/lys2Δ0 met17Δ0/MET17 ura3Δ0/ura3Δ0 ade2Δ::HIS3*/ ade2Δ::HIS3* TAF3$^{1-270}$-HA(KanMX6)/ TAF3 | This study |
| ByK725 | ByK533 and ByK484 | MATa/α his3Δ1/his3Δ1 leu2Δ0/leu2Δ0 LYS2/lys2Δ0 met17Δ0/MET17 ura3Δ0/ura3Δ0 ade2Δ::HIS3*/ ade2Δ::HIS3* psd2Δ::KanMX6/PSD2 dpl1Δ::NAT /DPL1 | This study |
| ByK726 | ByK533 and ByK484 | MATa/α his3Δ1/his3Δ1 leu2Δ0/leu2Δ0 LYS2/lys2Δ0 met17Δ0/MET17 ura3Δ0/ura3Δ0 ade2Δ::HIS3*/ ade2Δ::HIS3* psd2Δ::KanMX6/PSD2 dpl1Δ::NAT /DPL1 | This study |
| ByK739 | ByK725 | MATa/α his3Δ1/his3Δ1 leu2Δ0/leu2Δ0 LYS2/lys2Δ0 met17Δ0/MET17 ura3Δ0/ura3Δ0 ade2Δ::HIS3*/ ade2Δ::HIS3* psd2Δ::KanMX6/PSD2 dpl1Δ::NAT /DPL1 cdc10Δ::URA3/CDC10 | This study |
| ByK740 | ByK726 | MATa/α his3Δ1/his3Δ1 leu2Δ0/leu2Δ0 LYS2/lys2Δ0 met17Δ0/MET17 ura3Δ0/ura3Δ0 ade2Δ::HIS3*/ ade2Δ::HIS3* psd2Δ::KanMX6/PSD2 dpl1Δ::NAT /DPL1 cdc10Δ::URA3/CDC10 | This study |
| ByK741 | ByK726 | MATa/α his3Δ1/his3Δ1 leu2Δ0/leu2Δ0 LYS2/lys2Δ0 met17Δ0/MET17 ura3Δ0/ura3Δ0 ade2Δ::HIS3*/ ade2Δ::HIS3* psd2Δ::KanMX6/PSD2 dpl1Δ::NAT /DPL1 cdc10Δ::URA3/CDC10 | This study |
| YJM3916 | ByK352 | MATa his3Δ1 leu2Δ0 met17Δ0 ura3Δ0 ade2Δ::HIS3* YEN1$^{on}$ | This study |
| YL516 | BY4741/ BY4742 | MATa his3Δ1 leu2Δ0 ura3Δ0 | *Binda et al. (2009)* |
| MB32 | YL516 | MATa his3Δ1 leu2Δ0 ura3Δ0 gtr1Δ::kanMX | *Binda et al. (2009)* |
| RKH106 | YL516 | MATa his3Δ1 leu2Δ0 ura3Δ0 pib2Δ::kanMX | This study |
| RKH241 | MB32 | MATa his3Δ1 leu2Δ0 ura3Δ0 gtr1Δ::kanMX gtr2Δ::hphMX4 | This study |
| NMY51 | | his3Δ200 trp1-901 leu2-3,112 ade2 LYS::(lexAop)4-HIS3 ura3::(lexAop)8- lacZ ade2::(lexAop)8-ADE2 GAL4 | Dualsystems Biotech AG |

*\*ADE2 deleted −56 before ATG +62 after STOP with PCR primers #6 and #7 on pFA6a-His3MX6.*

adenine prototrophy to *ade2Δ* cells unless the Ac transposase excises the MiniDS transposon, and repairs the *ADE2* gene.

Deletion strains were generated by PCR-mediated gene replacement using the Longtine toolbox for *KanMX6* and *HIS3* replacement (*Longtine et al., 1998*) and the Janke toolbox for *NATnt2* (*Janke et al., 2004*), with primers listed in *Table 3*. Strain YJM3916 carrying *YEN1$^{ON}$* at the endogenous locus was generated using the *delitto perfetto* method (*Storici and Resnick, 2003*).

**Table 3.** Oligonucleotides used in this study.

| # | Original name | Sequence | Purpose |
|---|---|---|---|
| 1 | P5_MiniDs | AATGATACGGCGACCACCGAGATCTACtccgtcccgcaagttaaata | amplify library |
| 2 | MiniDs_P7 | CAAGCAGAAGACGGCATACGAGATacgaaaacgaacgggataaa | amplify library |
| 3 | 688_minidsSEQ1210 | tttaccgaccgttaccgaccgttttcatcccta | sequence library |
| 4 | ADE2Fwd | GGTTCGAGCTCCCTTTTGATGCGGAATTGAC | clone *ADE2* MiniDS |
| 5 | ADE2Rev | GACCTGAGCTCTTACTGGATATGTATGTATG | clone *ADE2* MiniDS |
| 6 | Ade2PriFwd | GTATAAATTGGTGCGTAAAATCGTTGGATCTCTCTTCTAAcggatccccgggttaattaa | delete *ADE2* |
| 7 | Ade2PriRev | TATGTATGAAGTCCACATTTGATGTAATCATAACAAAGCCgaattcgagctcgtttaaac | delete *ADE2* |
| 8 | Dpl1_Janke_S1 | AGCAAGTAGGCTAGCTTCTGTAAAGGGATTTTTCCATCTAATACAcgtacgctgcaggtcgac | delete *DPL1* |
| 9 | Dpl1_Janke_S2 | GCACTCTCGTTCTTTAAATTATGTATGAGATTTGATTCTATATAGatcgatgaattcgagctcg | delete *DPL1* |
| 10 | Psd2_pringle_F | GATGCTGTATCAATTGGTAAAGAATCCTCGATTTTCAGGAGCATCCAACGcgtacgctgcaggtcgac | delete *PSD2* |
| 11 | Psd2_pringle_R | CTTGTTTGTACACGCTATAGTCTATAATAAAGTCTGAGGGAGATTGTTCATGatcgatgaattcgagctcg | delete *PSD2* |
| 12 | TAF3_R1 | TGGATGAGATAATGACGAAAGAAAATGCAGAAATGTCGTTgaattcgagctcgtttaaac | *TAF3* partial deletion |
| 13 | TAF3_aa90_F2 | AGGTATTGTTAAGCCTACGAACGTTCTGGATGTCTATGATcggatccccgggttaattaa | TAF3 partial deletion |
| 14 | Taf3_Fwd | GGCAAGATGTGATCAGGACG | check *TAF3* partial deletion |
| 15 | Taf3_Rev | TCTTGAAGAAGCGAAAGTACACT | check *TAF3* partial deletion |
| 16 | TAF3_R1 | TGGATGAGATAATGACGAAAGAAAATGCAGAAATGTCGTTgaattcgagctcgtttaaac | *TAF3* complete deletion |
| 17 | TAF3_aa1_F1 | GAAAACAGCGATATCTTTGGGTCAATAGAGTTCCTCTGCTtgaggcgcgccacttctaaa | *TAF3* complete deletion |
| 18 | PRP45_R1 | ACTCAAGCACAAGAATGCTTTGTTTTCCTAGTGCTCATCCTGGGCgaattcgagctcgtttaaac | *PRP45* partial deletion |
| 19 | PRP45_aa154_F2 | AACGACGAAGTCGTGCCTGTTCTCCATATGGATGGCAGCAATGATcggatccccgggttaattaa | *PRP45* partial deletion |
| 20 | PRP45_Fwd | AGGTTGTAGCACCCACAGAA | check *PRP45* partial deletion |
| 21 | PRP45_Rev | CAATCATCACACCTCAGCGA | check *PRP45* partial deletion |
| 22 | PRP45_R1 | ACTCAAGCACAAGAATGCTTTGTTTTCCTAGTGCTCATCCTGGGCgaattcgagctcgtttaaac | *PRP45* complete deletion |
| 23 | PRP45_aa1_F1 | GCTCTGAGCCGAGAGGACGTATCAGCAACCTCAACCAAATtgaggcgcgccacttctaaa | *PRP45* complete deletion |
| 24 | CDC10-Ura3_fwd | AAGGCCAAGCCCCACGGTTACTACAAGCACTCTATAAATATATTAtgacggtgaaaacctctgac | *CDC10* complete deletion |
| 25 | URA3-CDC10_rev | TTCTTAATAACATAAGATATATAATCACCACCATTCTTATGAGATtcctgatgcggtattttctcc | *CDC10* complete deletion |
| 26 | OJM370 | ATGGGTGTCTCACAAATATGGG | Amplify *YEN1* |
| 27 | OJM371 | TTCAATAGTGCTACTGCTATCAC | Amplify *YEN1* |
| 28 | OJM372 | TTCAATAGTGCTACTGCTATCACTGTCACAGGCTCAAACCGGTCGACTG TTCGTACGCTGCAGGTCGAC | *Delitto perfetto* on *YEN1* |
| 29 | OJM373 | ATGGGTGTCTCACAAATATGGGAATTTTTGAAGCCATATCTGCAAGATTCCCGCGCGTTGGCCGATTCAT | *Delitto perfetto* on *YEN1* |
| 30 | o3958 | gacggtatcgataagcttgatatcgGCGCTGGCATCTTTAATCTC | *PIB2* cloning |
| 31 | o3959 | actagtggatcccccgggctgcaggTGCTTGGATCCTTCTTGGTC | *PIB2* cloning |
| 32 | o3224 | TAATA CGACT CACTA TAGGG | various *PIB2* truncations |

*Table 3 continued on next page*

Michel *et al.* eLife 2017;6:e23570. DOI: 10.7554/eLife.23570

*Table 3 continued*

| # | Original name | Sequence | Purpose |
|---|---|---|---|
| 33 | o3225 | ATTAA CCCTC ACTAA AGGGA A | various *PIB2* truncations |
| 34 | o4034 | atctagttcagggttcgacattctggtctccactac | *PIB2*$^{165-635}$ truncation |
| 35 | o4010 | gtagtggagaccagaatgtcgaaccctgaactagat | *PIB2*$^{165-635}$ truncation |
| 36 | o4012 | tagtggagaccagaatgttaccgcagcctgct | *PIB2*$^{304-635}$ truncation |
| 37 | o4035 | tcaaattagaactagcattcattctggtctccactacaactgtg | *PIB2*$^{221-635}$ truncation |
| 38 | o4011 | cacagttgtagtggagaccagaatgaatgctagttctaatttga | *PIB2*$^{221-635}$ truncation |
| 39 | o4062 | atagttggtattaagttgattctcattctggtctccactacaactg | *PIB2*$^{426-635}$ truncation |
| 40 | o3996 | cagttgtagtggagaccagaatgagaatcaacttaataccaactat | *PIB2*$^{426-635}$ truncation |
| 41 | o4063 | cgtgtttgcgttatggttgtcgctgttcggaataga | *PIB2*$^{\Delta426-532}$ truncation |
| 42 | o3997 | tctattccgaacagcgacaaccataacgcaaacacg | *PIB2*$^{\Delta426-532}$ truncation |
| 43 | o4064 | cacagagccgataacactcgtggttgaaaggttctc | *PIB2*$^{\Delta533-620}$ truncation |
| 44 | o3998 | gagaacctttcaaccacgagtgttatcggctctgtg | *PIB2*$^{\Delta533-620}$ truncation |
| 45 | o4065 | gtctcgcaaaaaatgttcatcagcccaaaacatcattaccttct | *PIB2*$^{1-620}$ truncation |
| 46 | o3999 | agaaggtaatgatgttttgggctgatgaacatttttttgcgagac | *PIB2*$^{1-620}$ truncation |
| 47 | o1440 | GCTAGAGCGGCCATTACGGCCCCGGAGATTTATGGACCTC | *KOG1* cloning into pPR3N |
| 48 | o1442 | CGATCTCGGGCCGAGGCGGCCTCAAAAATAATCAATTCTCTCGTC | *KOG1* cloning into pPR3N |
| 49 | o3787 | GCTAGAGCGGCCATTACGGCC GAATTGTACAAATCTAGAACTAGT | cloning *PIB2* fragments into pCabWT* |
| 50 | o3788 | CGATCTCGGGCCGAGGCGGCCAA GAAACTACTCCAATTCCAGTTTGC | cloning *PIB2* fragments into pCabWT* |
| 51 | o3872 | CGATCTCGGGCCGAGGCGGCCAAGCCCAAAACATCATTACCTTCTTCT | cloning *PIB2* fragments into pCabWT* |
| 52 | o3871 | CGATCTCGGGCCGAGGCGGCCAAATCTTCGCCCTCCTCAACGT | cloning *PIB2* fragments into pCabWT* |
| 53 | o3870 | CGATCTCGGGCCGAGGCGGCCAAGTTGATTCTGTCGCTGTTCG | cloning *PIB2* fragments into pCabWT* |
| 54 | o3933 | GCTAGAGCGGCCATTACGGCCAGGAAGAAATTACGCAATTACTAC | cloning *PIB2* fragments into pCabWT* |
| 55 | o3934 | GCTAGAGCGGCCATTACGGCC AGTGTTATCGGCTCTGTGCC | cloning *PIB2* fragments into pCabWT* |
| 56 | o3868 | CGATCTCGGGCCGAGGCGGCCAAATTAGTGCTCGAAGCAGGCT | cloning *PIB2* fragments into pCabWT* |

*Table 3 continued on next page*

*Table 3 continued*

| # | Original name | Sequence | Purpose |
|---|---|---|---|
| 57 | o3867 | CGATCTCGGGCCGAGGCGGCCAAGTCATCCGTGAATGGCAACG | cloning *PIB2* fragments into pCabWT* |
| 58 | o3866 | CGATCTCGGGCCGAGGCGGCCAAGCCTGCCCCTGTTGAGCTCT | cloning *PIB2* fragments into pCabWT* |
| 59 | o3865 | CGATCTCGGGCCGAGGCGGCCAAGTCAGCACCGCTTTCCTCAT | cloning *PIB2* fragments into pCabWT* |

Oligonucleotides #1 and #2, ordered as PAGE-purified and lyophilized, are resuspended at 100 µM in water. Oligonucleotide #3, ordered as HPLC-purified and lyophilized, is resuspended at 100 µM in water and distributed into single-use aliquots.

## Library generation

*ade2Δ* strains were transformed with the pBK257 plasmid. These strains are phenotypically *ade-*, since the *ADE2* gene borne on the plasmid is interrupted by the MiniDs transposon. One liter of freshly prepared SD -Ura +2% Raffinose +0.2% Dextrose is inoculated with *ade2Δ* cells freshly transformed with pBK257, directly scraped off the transformation plates, at a final OD600 = 0.15. The culture is grown to saturation for 18 to 24 hr at 30°C. Cells are spun for 5 min at 600x g, 20°C, and resuspended in their supernatant at a final OD600 = 39. 200 µl of this resuspension are plated on ~250–300×8.5 cm plates containing 25 ml of SD +2% Galactose -Adenine using glass beads. Plates are incubated in closed but not sealed plastic bags for 3 weeks at 30°C. Clones in which transposon excision has led to the repair of the *ADE2* gene on pBK257 start to appear after 10–12 days. The density of clones on the plate reaches 150–200 colonies/cm$^2$, i.e. 8000-11000 colonies/plates after 3 weeks. All colonies are then scraped off the plates using minimal volume of either water or SD +2% Dextrose -Adenine, pooled, and used to inoculate a 2-liter SD +2% Dextrose -Adenine culture at a density of 2.5 10$^6$ cells/ml, which is allowed to grow to saturation. This step is used to dilute any remaining ade- cells, which represent about 20% of the total number of cells, and ensures that each transposition event is well represented. For example, reseeding a 2 10$^6$ clones library in 2L at a density of 2.5 10$^6$ cells/ml will ensure that each clone is represented by ((2.500.000 × 1000×2)*0.8)/ 2.000.000 = 2000 cells. The saturated culture is harvested by centrifugation (5 min, 1600x g), washed with ddH2O, then cell pellets are frozen as ~500 mg aliquots.

## Rapamycin treatment

Cells scraped off the plates were used to inoculate a 1-liter SD +2% Dextrose -Adenine culture at OD 0.08. After growing for 15 hr to OD 0.5, the culture was diluted to OD 0.1 in 500 ml SD +2% Dextrose -Adenine, treated with 10 nM (9.14 ng/ml) rapamycin (Sigma) and grown for 24 hr to OD 0.9. The culture was then diluted again to OD 0.1 in 500 ml SD +2% Dextrose -Adenine + 10 nM rapamycin. The treated culture was grown to saturation (OD 1.9), harvested by centrifugation and processed for genomic DNA extraction.

## Genomic DNA

A 500 mg cell pellet is resuspended with 500 µl Cell Breaking Buffer (2% Triton X-100, 1% SDS, 100 mM NaCl, 100 mM Tris-HCl pH8.0, 1 mM EDTA) and distributed in 280 µl aliquots. 200 µl Phenol: Chloroform:Isoamylalcool 25:25:1 and 300 µl 0.4–0.6 mm unwashed glass beads are added to each aliquot. Samples are vortexed for 10 min at 4°C using a Disruptor Genie from Scientific Industrial (US Patent 5,707,861). 200 µl TE are added to each lysate, which are then centrifuged for 5 min at 16100x g, 4°C. The upper layer (~400 µl) is transferred to a fresh tube, 2.5vol 100% EtOH are added and the sample mixed by inversion. DNA is pelleted for 5 min at 16100x g, 20°C. The supernatant is removed and the pellets resuspended in 200 µl RNAse A 250 µg/ml for 15 min at 55°C, 1000 rpm on a Thermomixer comfort (Eppendorf). 2.5 vol 100% EtOH and 0.1 vol NaOAc 3 M pH5.2 are added and the samples mixed by inversion. DNA is pelleted by centrifugation for 5 min at 16100x g, 20°C. The pellets are washed with 70% EtOH under the same conditions, the supernatant removed

**Table 4.** Plasmids used in this study.

| Name | Parent | Description | Reference |
|---|---|---|---|
| pBK257 | pWL80R_4x | CEN/*URA3*, carries MiniDs in *ADE2* and hyperactive Ac transposase under *GAL1* promoter | This study |
| pWL80R_4x | | CEN/*URA3*, carries hyperactive Ac transposase under *GAL1* promoter | *Lazarow et al. (2012)* |
| pCORE-UH | | *Delitto pefetto URA3* cassette | *Storici and Resnick (2003)* |
| pJM7 | | pENTRY-*YEN1*$^{ON}$ | This study |
| pRS413 | | *CEN/HIS3*, empty vector | *Sikorski and Hieter, 1989* |
| pRS415 | | *CEN/LEU2*, empty vector | *Sikorski and Hieter, 1989* |
| pRS416 | | *CEN/URA3*, empty vector | *Sikorski and Hieter, 1989* |
| p1822 | pRS413 | *CEN/HIS3, GTR1* | This study |
| p1451 | pRS415 | *CEN/LEU2, GTR2* | This study |
| p1821 | pRS413 | *CEN/HIS3, GTR1*$^{Q65L}$ | This study |
| p1452 | pRS415 | *CEN/LEU2, GTR2*$^{S23L}$ | This study |
| p3084 | pRS416 | *CEN/URA3, PIB2* | This study |
| p3099 | p3084 | *CEN/URA3, PIB2*$^{165-635}$ | This study |
| p3097 | p3084 | *CEN/URA3, PIB2*$^{304-635}$ | This study |
| p3101 | p3084 | *CEN/URA3, PIB2*$^{221-635}$ | This study |
| p3253 | pRS426 | *2 µ/URA3, PIB2* | This study |
| p3255 | pRS426 | *2 µ/URA3, PIB2*$^{165-635}$ | This study |
| p3163 | p3084 | *CEN/URA3, PIB2*$^{426-635}$ | This study |
| p3153 | p3084 | *CEN/URA3, PIB2*$^{\Delta 426-532}$ | This study |
| p3154 | p3084 | *CEN/URA3, PIB2*$^{\Delta 533-620}$ | This study |
| p3156 | p3084 | *CEN/URA3, PIB2*$^{1-620}$ | This study |
| pPR3N | | *2 µ/TRP1, NubG-HA* | Dualsystems Biotech AG |
| pCabWT | | *CEN/LEU2, Aβ-Cub-LexA-VP16* | Dualsystems Biotech AG |
| p3081 | pPR3N | *2 µ/TRP1, NubG-HA-KOG1* | This study |
| p2966 | pCabWT | *CEN/LEU2, Aβ-PIB2-Cub-LexA-VP16* | This study |
| p3002 | pCabWT | *CEN/LEU2, Aβ-PIB2*$^{1-620}$*-Cub-LexA-VP16* | This study |
| p3007 | pCabWT | *CEN/LEU2, Aβ-PIB2*$^{1-550}$*-Cub-LexA-VP16* | This study |
| p3001 | pCabWT | *CEN/LEU2, Aβ-PIB2*$^{1-428}$*-Cub-LexA-VP16* | This study |
| p3051 | pCabWT | *CEN/LEU2, Aβ-PIB2*$^{440-550}$*-Cub-LexA-VP16* | This study |
| p3054 | pCabWT | *CEN/LEU2, Aβ-PIB2*$^{556-620}$*-Cub-LexA-VP16* | This study |
| p3052 | pCabWT | *CEN/LEU2, Aβ-PIB2*$^{621-635}$*-Cub-LexA-VP16* | This study |
| p3000 | pCabWT | *CEN/LEU2, Aβ-PIB2*$^{1-312}$*-Cub-LexA-VP16* | This study |
| p2987 | pCabWT | *CEN/LEU2, Aβ-PIB2*$^{304-635}$*-Cub-LexA-VP16* | This study |
| p2999 | pCabWT | *CEN/LEU2, Aβ-PIB2*$^{1-162}$*-Cub-LexA-VP16* | This study |
| p2986 | pCabWT | *CEN/LEU2, Aβ-PIB2*$^{165-635}$*-Cub-LexA-VP16* | This study |
| p2998 | pCabWT | *CEN/LEU2, Aβ-PIB2*$^{1-101}$*-Cub-LexA-VP16* | This study |
| p2991 | pCabWT | *CEN/LEU2, Aβ-PIB2*$^{102-635}$*-Cub-LexA-VP16* | This study |
| p2997 | pCabWT | *CEN/LEU2, Aβ-PIB2*$^{1-49}$*-Cub-LexA-VP16* | This study |
| p2990 | pCabWT | *CEN/LEU2, Aβ-PIB2*$^{50-635}$*-Cub-LexA-VP16* | This study |

completely, and the pellets dried for 10 min at 37°C. The pellets are resuspended in a total volume of 100 µl water for 10 min at 55°C, 700 rpm on a Thermomixer comfort (Eppendorf).

DNA is run on a 0.6% 1X TBE agarose gel against a standard 1 kb GeneRuler, and quantified using Fiji. 500 mg cell pellet should yield 20–60 µg DNA.

## Library sequencing

Sequencing involves the following steps: (1) Digestion of genomic DNA with two four-cutter restriction enzymes, (2) ligase-mediated circularization of the DNA, (3) PCR of the transposon-genome junctions using outward-facing primers, (4) Illumina-sequencing of the combined PCR products.

2 × 2 µg of genomic DNA are digested in parallel in Non-Stick microfuge tubes (Ambion AM12450) with 50 units of DpnII (NEB #R0543L) and NlaIII (NEB #R0125L), in 50 µl for 16 hr at 37°C. The reactions are then heat inactivated at 65°C for 20 min and circularized in the same tube by ligation with 25 Weiss units T4 Ligase (Thermo Scientific #EL0011) for 6 hr at 22°C, in a volume of 400 µl. DNA is precipitated overnight or longer at −20°C in 0.3 M NaOAc pH5.2, 1 ml 100% EtOH, using 5 µg linear acrylamide (Ambion AM9520) as a carrier, then centrifuged for 20 min at 16100x g, 4°C. Pellets are washed with 1 ml 70% EtOH, for 20 min at 16100 x g, 20°C. After complete removal of the supernatant, pellets are dried for 10 min at 37°C. Each circularized DNA preparation is then resuspended in water and divided into 10 × 100 µl PCR reactions. Each 100 µl PCR reaction contains: 10 µl 10X Taq Buffer (500 mM Tris-HCl pH9.2, 22.5 mM MgCl2, 160 mM NH4SO4, 20% DMSO, 1% Triton X-100 – stored at −20°C), 200 µM dNTPs, 1 µM primer #1, 1 µM primer #2, 2.4 µl homemade Taq polymerase. PCR is performed in an MJ Research Peltier Thermal Cycler PTC-200 using the following conditions:

Block: calculated – 95°C 1 min, 35 × [95°C 30 s, 55°C 30 s, 72°C 3 min], 72°C 10 min.

The 2 × 10 PCR reactions are pooled into one NlaIII-digested pool and one DpnII-digested pool. 100 µl from each pool are purified using a PCR clean-up/gel extraction kit (Macherey-Nagel) according to the manufacturer protocol, with the following modifications. DNA is bound to the column for 30s at 3000x g; 30 µl of elution buffer (10 mM Tris-HCl pH8.5, 0.1% Tween) is applied to the column and incubated for 3 min, then spun for 1 min at 11000x g at 20°C. The eluate is reapplied to the column and a second elution is performed under the same conditions. Purified PCR products are quantified by absorbance at 260 nm. On a 1% agarose gel, the product runs as a smear from 250 bp to 1.2 kb, with highest density centered around 500 bp. The 867 bp size band present in the NlaIII-treated sample and the 465 bp size band present in the DpnII-treated sample correspond to untransposed pBK257. Equal amounts of DpnII- and NlaIII-digested DNA are pooled and sequenced using MiSeq v3 chemistry, according to manufacturer, adding 3.4 µl of 100 µM primer #3 into well 12 of the sequencing cartridge.

## Bioinformatics analyses

The fastq file generated is uploaded into the CLC genomics workbench, trimmed using adaptor sequences 'CATG' and 'GATC' (the recognition sites for NlaIII and DpnII, respectively), allowing two ambiguities and a quality limit of 0.05. The trimmed sequence is then aligned to the reference genome, using the following parameters (mismatch cost, 2; insertion and deletion costs, 3; length fraction, 1; similarity fraction, 0.95; non-specific match handling, ignore). The alignment is then exported as a BAM file, which is further processed in MatLab, using the *Source code 1*, to detect individual transposition events. The outputted bed file is uploaded to the UCSC genome browser. Yeast annotations were downloaded from the Saccharomyces Genome Database (SGD). To generate our list of essential genes, we used YeastMine and searched the SGD for genes for which the null mutant has an 'inviable' phenotype (*Balakrishnan et al., 2012*).

Volcano plots were computed as follows. Two sets of libraries were defined. For each gene and each library, the number of transposons per gene (tnpergene variable) was normalized to the total number of transposons mapped in the library. For each gene, the fold-change is calculated as the mean of the normalized number of transposons per gene in the experimental set, divided by that in the reference set. The *p*-value is computed using the Student's t-test by comparing, for each gene, the normalized number of transposons per gene for each library in the experimental and reference sets.

## Western blotting

Cells were grown to mid-log phase in synthetic minimal medium containing 0.5 g/L proline as a sole nitrogen source and stimulated with 3 mM glutamine for 2 min. Cells were treated with 6.7% w/v tri-chloroacetic acid (final concentration), pelleted, washed with 70% ethanol and then lyzed in urea buffer (50 mM Tris-HCl [pH 7.5], 5 mM EDTA, 6 M urea, 1% SDS, 0.1 mg/ml Pefabloc/phosphatase inhibitor mix). After disrupting cells with glass beads and incubating with Laemmli SDS sample buffer, samples were subjected to regular SDS-PAGE and immunoblotting. The phosphorylation level of Sch9-Thr[737] and the total amount of Sch9 were assessed using the phosphospecific anti-Sch9-pThr[737] and anti-Sch9 antibodies, respectively (*Péli-Gulli et al., 2015*).

## Split-ubiquitin yeast two-hybrid assay

The split-ubiquitin yeast two-hybrid system from Dualsystems Biotech AG was used following the manufacturer's instructions.

Pib2 fragments (full-length or truncated) and full-length Kog1 were cloned into pCabWT and pPR3N plasmids, respectively, and transformed into the strain NMY51 as indicated. Protein-protein interactions were detected as growth of the resultant strains on agar plates lacking adenine.

## Accession numbers

Sequencing data have been deposited at EMBL-EBI ArrayExpress: E-MTAB-4885.

## Acknowledgements

We thank Beat Christen for inspiring this work and expert insight on Tn-seq, Reinhard Kunze for kind gift of plasmid and strain, Asun Monfort Pineda and Anton Wutz for invaluable help with Illumina Sequencing, Jeremy Thorner, Karsten Weis, Jeffrey Tang, Judith Berman and Vladimir Gritsenko for helpful discussions, Christine Doderer for preliminary experiments, Alicia Smith for comments on the manuscript, and the Kornmann lab for comments and ideas. This work is supported by grants of the Swiss National Science Foundation (PP00P3_13365 to BK, 310030_166474 to CDV, 31003A_153058 and 155823 to JM), the European Research Council (337906-OrgaNet) to BK, and PK is supported by the Human Frontier Science Program Organization.

## Additional information

### Funding

| Funder | Grant reference number | Author |
| --- | --- | --- |
| European Commission | 337906-OrgaNet | Benoît Kornmann |
| Schweizerischer Nationalfonds zur Förderung der Wissenschaftlichen Forschung | PP00P3_13365 | Benoît Kornmann |
| Human Frontier Science Program | | Philipp Kimmig |
| Schweizerischer Nationalfonds zur Förderung der Wissenschaftlichen Forschung | 310030_166474 | Claudio De Virgilio |
| Schweizerischer Nationalfonds zur Förderung der Wissenschaftlichen Forschung | 31003A_153058 | Joao Matos |
| Schweizerischer Nationalfonds zur Förderung der Wissenschaftlichen Forschung | 155823 | Joao Matos |

The funders had no role in study design, data collection and interpretation, or the decision to submit the work for publication.

## Author contributions

AHM, Conceptualization, Data curation, Formal analysis, Supervision, Validation, Investigation, Methodology, Writing—original draft, Writing—review and editing; RH, MA, Conceptualization, Investigation, Methodology, Writing—review and editing; PK, Formal analysis, Investigation, Writing—review and editing; MP, Supervision, Funding acquisition, Project administration; JM, CDV, Conceptualization, Supervision, Funding acquisition, Project administration, Writing—review and editing; BK, Conceptualization, Data curation, Software, Formal analysis, Supervision, Funding acquisition, Visualization, Methodology, Writing—original draft, Project administration, Writing—review and editing

## Author ORCIDs

Matthias Peter, http://orcid.org/0000-0002-2160-6824
Joao Matos, http://orcid.org/0000-0002-3754-3709
Benoît Kornmann, http://orcid.org/0000-0002-6030-8555

# Additional files

## Supplementary files

• Supplementary file 1. Processed dataset containing (1) the position and number of reads for all transposons in each library (in the WIG format), Processed dataset - Dpl1del.wig
Processed dataset - Mmm1Del_Vps13D716H.wig
Processed dataset - Psd2Del_Dpl1del.wig
Processed dataset - V13D716H.wig
Processed dataset - WildType1.wig
Processed dataset - WildType2.wig
Processed dataset - WT_plus_rapamycin.wig
Processed dataset - Yen1$^{on}$.wig
summaries of the number of transposon and number of reads per gene, for all genes in each library (in the TXT format).
Processed dataset - Dpl1del_pergene.txt
Processed dataset - Mmm1Del_Vps13D716H_pergene.txt
Processed dataset - Psd2Del_Dpl1del_pergene.txt
Processed dataset - V13D716H_pergene.txt
Processed dataset - WildType1_pergene.txt
Processed dataset - WildType2_pergene.txt
Processed dataset - WT_plus_rapamycin_pergene.txt
Processed dataset - Yen1$^{on}$_pergene.txt

• Supplementary file 2. Table of genes appearing as essential in our analysis (i.e., their density of transposon is below 1/400 bp), but were not previously annotated as essential. The likely explanation for the low transposon density is written in column B for each gene.

• Supplementary file 3. Data computed to draw the volcano plots (*Figure 4—figure supplement 1*)

• Source code 1. MatLab Script 1.

• Source code 2. MatLab Script 2.

## Major datasets

The following dataset was generated:

| Author(s) | Year | Dataset title | Dataset URL | Database, license, and accessibility information |
|---|---|---|---|---|
| AH Michel, R Hata- | 2016 | Rapid High-Resolution Functional | http://www.ebi.ac.uk/ar- | Publicly available at |

keyama, P Kimmig, M Arter, M Peter, J Matos, CD Virgilio, B Kornmann | Mapping of Yeast Genomes by Saturated Transposition | rayexpress/experiments/E-MTAB-4885/ | the EMBL-EBI ArrayExpress Website (accession no: E-MTAB-4885)

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
