## [Decision Letter]

Thank you for submitting your article "Functional Mapping of Yeast Genomes by Saturated Transposition" for consideration by *eLife*. Your article has been reviewed by three peer reviewers, one of whom is a member of our Board of Reviewing Editors, and the evaluation has been overseen by Kevin Struhl as the Senior Editor. The reviewers have opted to remain anonymous.

The reviewers have discussed the reviews with one another and the Reviewing Editor has drafted this decision to help you prepare a revised submission.

As you can see from the consolidated review below, the general evaluation of your work is quite positive, and with suitable revision it would be appropriate for publication in *eLife*. However, there were a number of specific issues that were raised, that would need to be attended to beforehand.

The most critical issues that would require new experiments are elaborated below in points #1-2. Additional issues that would not require new experiments to address are in points #3-5. Following that are the three original reviews. The points raised in them do not need to be addressed in a point-by-point rebuttal but they are included to give you a broader sense of the full range of issues that were raised by the three reviewers, as this may be helpful to you in the event that you make further improvements or refinements to the SATAY method.

1) For a paper to be published in a high profile place like *eLife* (including, in cases such as this one, papers that are focused on methodology), there should be some novel biological insight reported. The insights described in the paper were, for the most part, confirming previously known connections. Given the ease of generating this data, the authors should report on a screen that revealed some new biology of significance. Because this is primarily a methodological paper, the analysis does not need to be particularly deep, but it should be sufficient to make plain that something new and interesting has been uncovered.

2) In the synthetic-lethal screens, the "hits" are barely separated from the background. This can most clearly be seen in Figure 4 for the hits below the diagonal. To show the interesting hits that are discussed in the text, the authors drew an expanded box. It is apparent from looking at this box that the 'hits' are not far from the bulk signal. This raises the question of how much intrinsic noise there is in these screens. For example, if one were to do a WT vs. WT screen (i.e. a biological replicate), would the 'spread' of the signal in the 2D plot be as great as shown in Figure 4? If one knows what one is looking for it is easy to pick the needles out of the haystack. But if one is doing a novel screen and doesn't know what to expect, how much faith should be placed in hits that are at the edge of the data cloud, as is the case for those in Figure 4? Performing a WT vs. WT control would shed light on this question.

3) The authors describe in many places how this approach is superior to other genetic approaches, especially arrayed based ones. However, no specific head-to-head comparison is performed to justify this claim. For example, how would the SATAY DPL1 screen compare to other array-based procedures? In the absence of specific comparisons, the authors need to significantly attenuate their claims throughout the manuscript regarding the superiority of their method.

4) The authors discuss the fact that several genes annotated as non-essential were found in their method to be depleted for transposition suggesting that they are essential (at least under the growth conditions used in the study). It would be useful for the authors to provide a table of all newly identified essential genes, if there are any, that previous methods may have missed due to accumulation of suppressor mutations or aneuploidy.

5) Although the authors provide a link for a searchable platform to retrieve data, It would be beneficial to the readers if simple and clear excel tables would be provided (e.g. as supplementary tables) stating only the "hits" or all genes enriched/depleted for transposons in the various genetic backgrounds that the authors have tested

Reviewer #1:

The authors describe a method for identifying transposon insertions that yield a scorable phenotype. The general approach is to induce transposition in a large population of cells that contain an inducible transposon, and then enrich for cells that exhibit a particular phenotype that is not exhibited by the initial population of wild type cells. The sites of transposon insertion in the enriched cells are then identified by high-throughput sequencing. Similar methods of transposon mutagenesis have been described in the past (e.g. Snyder). What is new here is (i) introducing deep sequencing into the workflow as a means to identify insertion sites, and (ii) the observation that insertions can be identified in intergenic regions and essential genes. In the case of essential genes, the insertions identify regions of the gene that are not essential for life but that yield a scorable phenotype when disrupted. Another point raised by the authors is that their method does not require robotics for dealing with arrayed libraries of knockout mutants. The method clearly has potential utility. The question is whether it represents a sufficient advance to merit publication in *eLife*. In my opinion, it would be important to address the two issues below prior to publication:

1) In the synthetic-lethal screens, the "hits" are barely separated from the background. This can most clearly be seen in Figure 4 for the hits below the diagonal. To show the interesting hits that are discussed in the text, the authors drew an expanded box. It is apparent from looking at this box that the 'hits' are not far from the bulk signal. This raises the question of how much intrinsic noise there is in these screens. For example, if one were to do a WT vs. WT screen (ie a biological replicate), would the 'spread' of the signal in the 2D plot be as great as shown in Figure 4? This seems like an important question to resolve, because of course if one knows what one is looking for it is easy to pick the needles out of the haystack. But if one is doing a novel screen and doesn't know what to expect how much faith could be placed in hits that are at the edge of the data cloud, as is the case for those in Figure 4? What would be useful here is (i) to see the WT vs. WT control, and (ii) to develop an algorithm that identifies hits in an automated manner and assigns a statistical significance to them.

2) It would appear that this method leads to identification of transposon hops in essential genes with relatively high frequency. The 3' insertions are easy to understand, the 5' insertions less so. The authors suggest that perhaps these arise from spurious transcription initiation events but this is speculative and not investigated. It might be useful for future users of this method if the authors were to dig into a few of these 5' insertions, to identify the mechanism(s) giving rise to these hits. This is especially true in this case because the primary value of the paper is as a methods paper. In that case it would be useful to have as deep an understanding of the method as possible.

Reviewer #2:

Michel and colleagues describe in this paper an elegant new way in budding yeast to carry out genetic screens using an approach called SATAY (Saturated Transposon Analysis in Yeast), which exploits systematic transposon mutagenesis coupled to high-throughput sequencing. This interesting protocol, which seemingly can be applied to other yeast species with ease, allows for genetic interaction mapping, drug target discovery, essential gene identification and protein domain analysis.

Interestingly, they show that inter-nucleosomal regions are preferred for transposon introduction as were pericentric regions (although they demonstrated that this was due to the constructs used for the screens). They also accurately identified the essential genes in the genome (under the conditions used); and could also identify key domains in a number of proteins, including Gal10, Taf3 and Prp45. They used the approach to carry out a screen using a specific point mutant that they had previously worked on and obtained expected results based on previous work and also carried out a screen using a DPL1 mutant and reported interactions that made sense with previous work. Finally, they used SATAY in the presence of Rapamycin to identify the isomerase Fpr1, as well as other genes involved in the TOR pathway. Overall, I am very positive and excited about this paper.

However, there are two main points that I feel would need to be addressed for me to support publication in *eLife*.

1) The authors describe in many places how this approach is superior to other genetic approaches, especially arrayed based ones. I would like to see how the interactions from this screening procedure compare to others using the same mutants. They carried out a screen with DPL1-how does the data derived from SATAY compare to other arrayed based procedures? I would like to see a few other screens and how the data compare. It may be worthwhile to have some gold standards and show via a ROC curve how this approach compares to other ones.

2) For a paper to be published in a high profile place like *eLife* (even a methodology paper), I feel there should be some novel biological insight reported. The insights described in the paper were, for the most part, confirming previously known connections. Given the ease of generating this data, I feel there should be a few more screens (with deletions, point mutants and/or drugs), with some novel biology derived from the data.

Reviewer #3:

The manuscript by Michel, Kimmig & Kornmann describes an innovative technology to rapidly and efficiently screen yeast for causal genomic loci. The approach, based on transposon mutagenesis followed by deep sequencing, takes advantage of the recent ease of using deep sequencing to circumvent most of the disadvantages of previous yeast screening technologies. Specifically, the new approach, termed SATAY, negates the use of arrayed libraries thus reducing the time and cost of previous screening strategies. I believe that this elegant method will be highly utilized by the yeast community. Since yeast is still the most powerful model organism for genetic screens, this technology is important and merits publication. Another potential application of this technology is its ability to map the 3D architecture of the genome with ease under a variety of conditions. In addition, the manuscript gives beautiful new insight to the domain architecture of essential genes.

---

## [Author Response]

*The most critical issues that would require new experiments are elaborated below in points #1-2. Additional issues that would not require new experiments to address are in points #3-5. Following that are the three original reviews. The points raised in them do not need to be addressed in a point-by-point rebuttal but they are included to give you a broader sense of the full range of issues that were raised by the three reviewers, as this may be helpful to you in the event that you make further improvements or refinements to the SATAY method.*

*1) For a paper to be published in a high profile place like eLife (including, in cases such as this one, papers that are focused on methodology), there should be some novel biological insight reported. The insights described in the paper were, for the most part, confirming previously known connections. Given the ease of generating this data, the authors should report on a screen that revealed some new biology of significance. Because this is primarily a methodological paper, the analysis does not need to be particularly deep, but it should be sufficient to make plain that something new and interesting has been uncovered.*

Our dataset uncovers over 300 domains among essential proteins. This is a unique dataset for a eukaryotic genome. We understand, however, that to demonstrate that our method is also useful for comparative library screening, we must show more than proof-of-principle and confirmation of previously known connections.

We have therefore included true biological findings from 2 screens that were part of the original manuscript and from an additional library.

1) One strength of our approach is to identify situations where genes insofar deemed essential become dispensable. We have identified two such cases in our screens and confirmed the interactions by classical tetrad analysis.

The first concerns the septin Cdc10. CDC10 is essential for growth at 30˚C but not at 25˚C. We find that cdc10 deletion lethal phenotype at 30˚C is partially alleviated by the deletion of DPL1. Dpl1 being is a negative regulator of sphingolipid accumulation, this result suggests that altering membrane composition renders Cdc10 dispensable, thus hinting at an important role of membrane lipids in septin ring assembly. These finding constitutes the new Figure 5, panels A-C.

The second concerns the essential replication factor Dna2. We generated a SATAY library in a strain expressing a constitutively active version of the Holliday-junction resolvase Yen1. We observe that DNA2 becomes dispensable in this strain, indicating that Yen1 could substitute entirely for Dna2 function, provided that it is constitutively active. This finding challenges previous functions attributed to both Yen1 and Dna2 and constitutes the new Figure 5, panels D-F.

2) Another unique strength of our approach is to unveil domain organization in proteins. We had identified Pib2 N-terminal truncations as causing a possible gain-of-function rapamycin resistance. We have investigated the phenomenon further. We find that Pib2 is an activator of the Target-of-rapamycin complex 1 (TORC1) and, guided by the transposon map, confirm that N-terminus pib2 truncations lead to semi-dominant rapamycin resistance. We originally assumed that the N-terminus negatively regulated Pib2 activity, possibly via an allosteric mechanism. Instead, further analyses showed that Pib2 possesses two independent antagonistic activities; one that stimulates and one that represses TORC1. N-terminal truncations of Pib2 activate TORC1 in a semi-dominant manner, while C-terminal truncations repress TORC1, also in a semi-dominant manner. Thus, our data unveil an entirely unanticipated architecture in Pib2, bearing two antagonistic activities. As the mechanisms for nutrient sensing by TORC1 are unclear, our data suggest that an upstream player can act both as an activator and a repressor of TORC1 depending on the conditions. It is, to the best of our knowledge, unheard of that a protein bears two antagonistic activities at each end of the polypeptide chain. These finding are described in the Figure 6, new panels C-H, and the new Figure 6—figure supplement 1.

*2) In the synthetic-lethal screens, the "hits" are barely separated from the background. This can most clearly be seen in Figure 4 for the hits below the diagonal. To show the interesting hits that are discussed in the text, the authors drew an expanded box. It is apparent from looking at this box that the 'hits' are not far from the bulk signal. This raises the question of how much intrinsic noise there is in these screens. For example, if one were to do a WT vs. WT screen (i.e. a biological replicate), would the 'spread' of the signal in the 2D plot be as great as shown in Figure 4? If one knows what one is looking for it is easy to pick the needles out of the haystack. But if one is doing a novel screen and doesn't know what to expect, how much faith should be placed in hits that are at the edge of the data cloud, as is the case for those in Figure 4? Performing a WT vs. WT control would shed light on this question.*

Our original submission contained two wild-type libraries, and their pairwise comparison indeed showed a significant spread (not shown). As pointed by reviewer #1, the fact that noise may blur the signal from the datasets is indeed an important issue that was not addressed in our previous manuscript.

The issue is particularly important when comparing libraries in a pairwise fashion, as we do in Figure 4. This issue however withers when a library is not compared in a pairwise fashion to another library, but is compared to a pool of other libraries. For instance, a hit can be confidently identified by manually verifying it in the genome browser and comparing transposon accumulations across all libraries. This is exemplified in Figure 4. We understand of course that this manual step is not satisfactory. We have resolved the issue by computing a metrics that allows to compare a library against a set of libraries, or sets of libraries against each other, and devised a way to graphically display the comparisons (Figure 4—figure supplement 1). We calculated the average transposon number per gene in each library set and determined a fold change across sets. We associated the fold change with a confidence score (based on a Student’s t-test), that reflects the spread in transposon number within each set. We generated volcano plots in which genes are plotted according to the fold change (x-axis) and the confidence score (y-axis). Using this approach, the hits are well separated and noise is efficiently cancelled out.

Moreover, this approach is very versatile. One library can be compared against all other libraries available, or a chosen set of libraries against another chosen set, depending on the question asked.

*3) The authors describe in many places how this approach is superior to other genetic approaches, especially arrayed based ones. However, no specific head-to-head comparison is performed to justify this claim. For example, how would the SATAY DPL1 screen compare to other array-based procedures? In the absence of specific comparisons, the authors need to significantly attenuate their claims throughout the manuscript regarding the superiority of their method.*

We understand this concern and have soften and clarified our wording; we do not know and do not claim that our approach is superior to library-based screens in terms of quantativity, sensitivity, robustness or reproducibility. That said, our approach bears a set of specific advantages over library- based screens as it allows the discovery of protein domains, the screening of the coding and non- coding genomes, and the discovery of a diversity of alleles. It is amenable to complex genetic backgrounds, and requires only simple equipment. Our approach is advantageous over classical mutagenesis in that it is much more comprehensive.

We have removed several sentences where we made possibly misleading statements comparing our approach to library-based screens. We have added full paragraphs in the Results and the Discussion sections to indicate the potential pitfalls in terms of signal-to-noise ratio (see above), and ways to circumvent them.

While it is, in principle, a great idea to compare our *DPL1* library with published library-based screens, *DPL1* unfortunately does not genetically interact with many high-confidence hits, neither in our data, nor in published datasets. Its best negative-genetic interactor, according to Boone et al. (thecellmap.org) is *LCB3* (score -0.851). This gene makes a lot of sense, since it is, like *DPL1*, involved in the turnover of dihydro- and phytosphingosin-phosphate. We also found it among our best hits. The second best hit of Boone et al. is *RAD1* with a score (-0.479) that is already half of LCB3’s. *RAD1* is a DNA repair protein, and a genetic interaction with LCB3 is difficult to make sense of. It is quite likely a false positive. RAD1 does not show a particular genetic interaction in our libraries. So specific head-to-head comparison is difficult.

*4) The authors discuss the fact that several genes annotated as non-essential were found in their method to be depleted for transposition suggesting that they are essential (at least under the growth conditions used in the study). It would be useful for the authors to provide a table of all newly identified essential genes, if there are any, that previous methods may have missed due to accumulation of suppressor mutations or aneuploidy.*

We filtered our dataset to find genes with lowest transposon density and included them in such a list ([Supplementary-material SD2-data]). Genes present in this list can be classified as follow:

1) Genes that are repeated. Our mapping strategy does not map reads that can be ambiguously mapped to more than one place in the genome. Therefore repeated genes appear as transposon-free,

2) Auxotrophy genes. We grow cell on synthetic medium, therefore many genes that confer prototrophy to a chemical that is absent from our medium appear as tansposon-free.

3) Galactose pathway genes. Since we grow cells on galactose, they require an intact galactose pathway.

4) Clearly misannotated genes. Some genes are known to be essential, yet they do not bear the “null mutant: inviable” GO-term that we used to populate the list of essential genes.

5) ORFs that overlap with essential genes. Several annotated dubious ORFs actually overlap with essential genes on most of their length.

6) New essential genes. The genes that do not fall into any of the above category represent potential new essential genes.

However we would like to raise attention to the intrinsic problems of classifying genes as either “essential” or “non-essential”. This problem is not specific to our approach, and we suspect that many genes lie in a grey zone between “essential” and “non-essential”, depending on the method used to address their “essentiality”. In our case, a mutation that slows down yeast growth by a factor of, say, 100 will cause a transposon-free area on our maps, yet the gene might not be considered as essential sensu stricto. We suspect that many genes that have a low transposon density are not essential sensu stricto, but their mutation renders cells extremely sick.

Yet, we note that, as anticipated by reviewer #3, 12 duplicated ribosomal protein genes appear in the list of new essential genes (e.g. RPL11A). Such genes are likely to be compensated for in deletion libraries, for instance by duplicating the chromosome bearing the second copy of the gene (e.g. RPL11B). The fitness as assessed by our method is indeed likely to reflect the fitness of an unadapted mutant strain.

*5) Although the authors provide a link for a searchable platform to retrieve data, It would be beneficial to the readers if simple and clear excel tables would be provided (e.g. as supplementary tables) stating only the "hits" or all genes enriched/depleted for transposons in the various genetic backgrounds that the authors have tested.*

We have added the requested full excel tables with scores used to compute the volcano plots ([Supplementary-material SD3-data], see above), as well as the transposon per genes and reads per genes for each library (Processed Dataset).